# HexGen-3: A Fully Disaggregated LLM Serving Framework with Fine-Grained Heterogeneous Resource Autoscaling

Youhe Jiang [* 1]   Wenshuang Li [* 1]   You Peng [1]   Jintao Zhang [2]   Ran Yan [1]   Jianfei Chen [2]   Xu Han [2]
Fangcheng Fu [3]   Binhang Yuan [1]

## Abstract

The operational cost of serving large language models remains prohibitively high, largely due to extreme *workload heterogeneity* in production traffic. We observe that combining disaggregated inference with resource autoscaling enables *fine-grained* resource adjustment, allowing inference phases and operations to scale independently based on their specific bottlenecks. Building on this insight, we propose HexGen-3, a cost-effective LLM serving framework that leverages a fully disaggregated inference architecture and heterogeneous resource autoscaling. HexGen-3 introduces two key components: (**i**) A *hierarchical scheduling framework* that jointly optimizes resource allocation and parallelism configuration for *any* given resource provisioning, and (**ii**) an *autoscaling framework* that dynamically adjusts resources and triggers deployment rescheduling in response to workload fluctuations. Experiments comparing HexGen-3 against state-of-the-art LLM serving systems demonstrate up to 60% (on average 46.5%) improvement in per-cost throughput under static resource provisioning, and up to 78.3% (on average 55.1%) improvement with autoscaling enabled under dynamic workloads.

## 1. Introduction

Large Language Models (LLMs) have revolutionized artificial intelligence, creating an urgent demand for scalable inference serving systems. However, the operational cost of serving LLMs with billions of parameters (e.g., the DeepSeek-v3 (Guo et al., 2025) and Llama-4 (Grattafiori

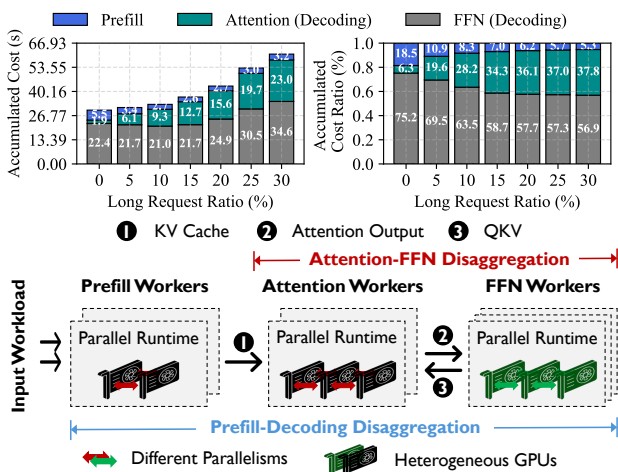

*Figure 1.* **Upper:** Attention and FFN loads change according to different long ($\approx$10k tokens (An et al., 2024)) versus short ($\approx$1k tokens (Patel et al., 2024)) request mix ratios. When incorporating 30% long requests, the prefill cost decreases 1.7$\times$ (a 3.5$\times$ decrease in ratio); the attention cost surges 12$\times$ (a 6$\times$ increase in ratio); while the FFN cost rises only 1.5$\times$ (a 1.3$\times$ decrease in ratio). **Lower:** Exampled architecture of a fully disaggregated system.

et al., 2024)) remains prohibitively high. A critical factor driving these inefficiencies is the *workload heterogeneity* inherent in real-world production traffic, such as the diverse mix of long versus short requests (Agrawal et al., 2024; Jiang et al., 2025a; Zhao et al., 2024b) and the varying balance of compute versus memory needs (Patel et al., 2024; Zhong et al., 2024). In this paper, we address this challenge by *leveraging both a fully disaggregated inference architecture and fine-grained heterogeneous resource autoscaling.*

Disaggregated inference has emerged as a promising architectural paradigm for LLM serving. LLM inference consists of two distinct phases: The *prefill* phase, which processes the entire input prompt in parallel and is compute-bound, and the *decoding* phase, which generates tokens autoregressively and is memory-bandwidth-bound due to repeated model weight loading. By separating these phases onto dedicated GPU pools, prefill-decoding disaggregation eliminates interference between their conflicting resource demands, allowing prefill workers to maximize GPU compute utilization while decoding workers can be optimized

[*]Equal contribution  [1]The Hong Kong University of Science and Technology, Hong Kong SAR, China [2]Tsinghua University, Beijing, China [3]Shanghai Jiao Tong University, Shanghai, China. Correspondence to: Binhang Yuan <biyuan@ust.hk>.

*Proceedings of the 43rd International Conference on Machine Learning*, Seoul, South Korea. PMLR 306, 2026. Copyright 2026 by the author(s).

for memory throughput. Furthermore, within the decoding phase, the *attention* operation (memory-bound, scaling with sequence length) exhibits different characteristics from the *Feed-Forward Network (FFN)* operation (compute-bound, fixed cost per token). Attention-FFN disaggregation (AFD) further separates these operations, enabling independent scaling and hardware matching for each. An example of a fully disaggregated architecture is illustrated in Figure 1. Recent studies (Zhong et al., 2024; Patel et al., 2024; Qin et al., 2024; Zhu et al., 2025) have demonstrated the effectiveness of disaggregated inference in production environments.

Complementary to disaggregation, resource autoscaling addresses the challenge of fluctuating workload demands. This technique dynamically provisions computational resources during periods of high traffic and releases them when demand subsides, thereby maintaining service quality while minimizing operational costs. Several studies (Lai et al., 2025; Li et al., 2025) have investigated autoscaling strategies tailored to LLM serving systems.

A core observation driving our work is that when combined with a disaggregated architecture, autoscaling enables *fine-grained* resource adjustment—allowing the prefill and decoding phases, as well as the attention and FFN operations, to scale independently based on their specific bottlenecks rather than scaling the entire system monolithically. As illustrated in Figure 1, workload heterogeneity (e.g., varying mix ratios of long versus short requests) leads to divergent prefill, attention, and FFN loads over time, resulting in varying resource requirements that demand independent scaling.

Hardware heterogeneity adds further complexity and opportunity to this landscape. Commercial and private computing platforms host a diverse range of GPU types, each offering different ratios of compute capability (FLOPs) to memory bandwidth. Previous works (Jiang et al., 2023; 2025a;d;b; Miao et al., 2024; Mei et al., 2025; Mao et al., 2025; Zhu et al., 2025) have demonstrated that strategic utilization of heterogeneous GPU resources significantly enhances cost-efficiency in LLM serving, making *heterogeneity-aware* scheduling essential.

Motivated by these observations, we propose HEXGEN-3, a framework that leverages a fully disaggregated inference architecture, resource autoscaling, and heterogeneous GPU resources to boost the cost-efficiency of LLM serving. Our contributions are as follows:

**Contribution 1: Scheduling framework.** We formulate the scheduling problem for fully disaggregated model deployment on heterogeneous GPU resources as a constrained optimization problem. To solve this problem efficiently, we propose a *hierarchical scheduling framework* that comprises (**i**) a *global scheduler* for optimizing resource allocation and (**ii**) a *local scheduler* for optimizing parallelism configura-

tion across different inference phases and operations. This framework is designed to operate on *any* given resource provisioning, enabling it to serve as the optimization core that can be invoked whenever resources change.

**Contribution 2: Autoscaling framework.** Building upon the scheduling framework, we design a *novel autoscaling framework* that comprises (**i**) an *autoscaling metric* for triggering scaling decisions and (**ii**) an *autoscaling method* for determining the new resource provisioning and model deployment strategy. It enables *fine-grained* and *independent* scaling of different inference phases and operations, allowing the system to adapt efficiently to workload fluctuations and enhance the cost-efficiency of LLM serving.

**Contribution 3: Implementation and evaluation.** We implement HEXGEN-3 with high-performance realizations of prefill-decoding and attention-FFN disaggregation, and conduct comprehensive evaluations against state-of-the-art LLM serving systems. Experiments comparing HEXGEN-3 against state-of-the-art LLM serving systems demonstrate up to 60% (on average 46.5%) improvement in per-cost throughput under static resource provisioning, and up to 78.3% (on average 55.1%) improvement with autoscaling enabled under dynamic workloads.

## 2. Background and Related Work

**Disaggregated LLM serving.** Since different inference phases/operations exhibit significantly different computational characteristics, recent efforts (Zhong et al., 2024; Patel et al., 2024; Chen et al., 2024) propose utilizing separate hardware resources for each phase/operation to avoid performance interference. In a fully disaggregated LLM serving architecture, the system deploys three types of specialized model replicas (workers): Prefill, attention, and FFN workers, with inter-worker communication to exchange KV cache and intermediate activations. However, existing phase splitting frameworks either (**i**) fail to *fully* disaggregate inference phases with distinct computational characteristics (Hu et al., 2024; Li et al., 2025) or (**ii**) enable deployment scheduling based on *static* resource provisioning (Qin et al., 2024; Zhu et al., 2025; Jiang et al., 2025b), leading to inefficient resource utilization under dynamic workloads.

**Heterogeneous LLM serving.** Heterogeneous LLM serving refers to systems that utilize a mix of GPU types with varying hardware characteristics—such as different computational and memory capacities—to optimize resource utilization and cost-efficiency (Miao et al., 2024; Mao et al., 2025; Jiang et al., 2025a;b;d; 2023; Mei et al., 2025). Existing works demonstrate that strategic utilization of heterogeneous resources significantly enhances the cost-efficiency of LLM serving. HEXGEN-3 advances this paradigm by optimizing heterogeneous model deployment within a *fully* disaggregated architecture that separates prefill, attention,

and FFN workers, thereby maximizing the utilization of heterogeneous computational resources.

**Autoscaling in LLM serving.** Autoscaling dynamically adjusts computational resources in response to workload fluctuations, maintaining quality of service while minimizing operational costs in LLM serving (Fu et al., 2024; Sun et al., 2024; Lai et al., 2025). Recent work has addressed key challenges in this domain: HeteroScale (Li et al., 2025) performs coordinated autoscaling through large-scale empirical analysis of production metrics, scaling resources based on distinct prefill and decoding characteristics. In contrast, HEXGEN-3 employs *fine-grained* autoscaling that decouples attention and FFN worker scaling within each phase, effectively mitigating resource stranding from workload heterogeneity across model components. Extended related work is provided in Appendix B.

## 3. System Overview

HEXGEN-3 comprises two synergistic components: A *scheduling framework* (§4) and an *autoscaling framework* (§5). Given a **static** resource provisioning and workload profile, the scheduling framework determines the optimal deployment strategy through two hierarchical stages: (**i**) A global scheduler that optimizes resource allocation across worker types (§4.2.1), and (**ii**) a local scheduler that optimizes parallelism configuration within each worker group (§4.2.2). Given **dynamic** workload fluctuations, the autoscaling framework orchestrates timely deployment adjustments through two coordinated mechanisms: (**i**) A monitoring module that continuously tracks real-time service metrics to detect workload shifts (§5.1), and (**ii**) a scaling module that triggers resource provisioning changes (§5.2.1) and reinvokes the scheduling framework for deployment rescheduling (§5.2.2). In essence, the scheduling framework serves as the *optimization core* for **static** resource-workload pairs, while the autoscaling framework serves as the *adaptation layer* for **dynamic** workload fluctuations—together ensuring cost-efficient serving under production traffic.

## 4. Scheduling Framework

The scheduling framework is responsible for optimizing the model deployment strategy for *any* given static resource provisioning and workload profile, enabling it to serve as the optimization core that can be invoked by the autoscaling framework (§5) whenever resources or workloads change.

### 4.1. Scheduling Formulation

Given a heterogeneous GPU cluster and incoming workload demands, we aim to determine the optimal model deployment strategy that maximizes overall system throughput. This strategy encompasses (**i**) resource allocation and (**ii**) parallelism configuration across prefill, attention (decoding), and FFN (decoding) workers.

**Decision variables.** Consider a heterogeneous GPU cluster $\mathbf{D} = \{d_1, d_2, \ldots, d_N\}$, where $d_n \in \mathbb{Z}_{\geq 0}$ denotes the total number of available GPUs of type $n \in \{1, 2, \ldots, N\}$. The system comprises three distinct worker types indexed by the set $\mathbf{T} = \{\text{pre}, \text{attn}, \text{ffn}\}$, representing prefill workers, attention workers, and FFN workers. The optimization problem seeks to determine two key decision variables:

- **Resource allocation matrix:** Let $\mathbf{A} \in \mathbb{Z}_{\geq 0}^{3 \times N}$ denote the resource allocation matrix. For each worker type $t \in \mathbf{T}$, we define the corresponding row vector as $\mathbf{A}_t = \langle a_{t,1}, \ldots, a_{t,N} \rangle$, where the element $a_{t,n}$ represents the number of GPUs of type $n$ allocated to worker type $t$.

- **Parallelism configuration:** Let $\mathbf{P} = \{\mathbf{P}_{\text{pre}}, \mathbf{P}_{\text{attn}}, \mathbf{P}_{\text{ffn}}\}$ denote the parallelism configuration. For each worker type $t \in \mathbf{T}$, we define the corresponding configuration vector as $\mathbf{P}_t = \langle p_{t,1}, \ldots, p_{t,N} \rangle$, where the element $p_{t,n}$ specifies the parallelism strategy (e.g., a tuple $p_{t,n} = \langle \delta_{\text{DP}}, \delta_{\text{TP}}, \delta_{\text{EP}} \rangle$ denoting data, tensor, and expert parallelism degrees) for the group of $a_{t,n}$ GPUs assigned to worker type $t$ with GPU type $n$.

**Optimization problem.** The system functions as a sequential pipeline (as shown in Figure 1) : Prefill $\rightarrow$ attention $\rightarrow$ FFN, where the global throughput (in req/s) is constrained by the bottleneck worker type. We formulate the deployment strategy as a constrained optimization problem aimed at maximizing the minimum effective throughput across all worker types: $\max_{\mathbf{A},\mathbf{P}} \min_{t \in \mathbf{T}} \Theta_t(\mathbf{W}, \mathbf{A}_t, \mathbf{P}_t)$, subject to $\sum_{t \in \mathbf{T}} a_{t,n} \leq d_n, \forall n \in \{1, \ldots, N\}$, where $\Theta_t(\cdot)$ denotes the estimated throughput for worker type $t \in \mathbf{T}$, given the incoming workload profile $\mathbf{W}$[1], resource allocation $\mathbf{A}_t$, and parallelism configuration $\mathbf{P}_t$. The objective seeks to elevate the performance of the slowest worker type, while the constraint ensures that the number of GPUs allocated across all worker types does not exceed the cluster's capacity.

### 4.2. Scheduling Solver

The optimization problem is non-linear and NP-hard. Thus, we decouple the decision variables and adopt a hierarchical scheduling framework (workflow shown in Algorithm 1): We utilize a global scheduler (§4.2.1) to explore different resource allocations, and a local scheduler for obtaining the optimal parallelism configuration (§4.2.2).

#### 4.2.1. GLOBAL SCHEDULER

We utilize a global scheduler to search for the resource allocation matrix $\mathbf{A}$, formulating this search process as a finite-horizon Markov Decision Process (MDP) (Puterman, 2014). In this framework, the global scheduler acts as an intelligent agent, sequentially refining the allocation by transferring resources between worker types to resolve

---

[1]Formally, $\mathbf{W}$ comprises the request arrival rate $\lambda$, and probability distributions for input token length $\mu_{\text{in}}$ and output token length $\mu_{\text{out}}$, derived from a sliding window of recent traffic.

**Algorithm 1** Hierarchical Scheduling Workflow

1: **Input:** $\mathbf{W}$: workload profile; $\mathbf{A}_0$: initial allocation; $\mathcal{H}$: stability threshold; $\psi_0$: initial temperature; DECAY$(\cdot)$: cooling function in Simulated Annealing (SA) process
2: **Intermediate:** $k$: iteration counter; $\psi_k$: current temperature; $\boldsymbol{\Theta}$: throughput profile; $\mathbf{P}$: parallelism configuration; $\mathbf{A}$: current allocation; $\mathbf{A}'$: candidate allocation
3: *// Initialize variables*
4: $\mathbf{A} \leftarrow \mathbf{A}_0; \quad k \leftarrow 0; \quad \psi_k \leftarrow \psi_0$
5: **while** true **do**
6:    *// Optimize parallelism configuration (§4.2.2)*
7:    $(\boldsymbol{\Theta}, \mathbf{P}) \leftarrow$ LOCALSCHEDULER$(\mathbf{W}, \mathbf{A})$
8:    *// Optimize resource allocation (§4.2.1)*
9:    $\mathbf{A}' \leftarrow$ GLOBALSCHEDULER$(\mathbf{A}, \boldsymbol{\Theta}, \psi_k)$
10:   *// Terminate upon convergence*
11:   **if** $\mathbf{A}$ is stable for $\mathcal{H}$ iterations **then**
12:     **break**
13:   **end if**
14:   *// Update allocation and SA temperature*
15:   $\mathbf{A} \leftarrow \mathbf{A}'; \quad \psi_{k+1} \leftarrow$ DECAY$(\psi_k); \quad k \leftarrow k + 1$
16: **end while**
17: **Return** $(\mathbf{A}, \mathbf{P})$

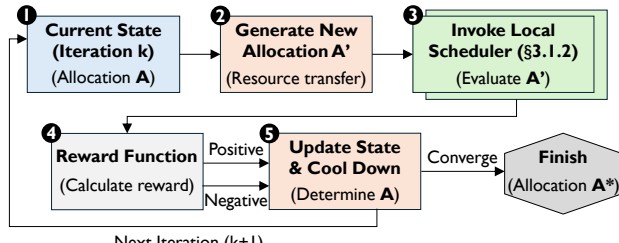

*Figure 2.* Global scheduler/Simulated annealing optimization loop.

$\mathbf{A}_0$, where available GPUs are distributed evenly among worker types. At each iteration k, SA generates a candidate action $u_k$ according to the guided generation policy to transition from state $s_k$ to $s_{k+1}$, thereby generating a new allocation $\mathbf{A}'$. The transition is accepted unconditionally if it yields a positive reward ($r_k > 0$, indicating throughput improvement) and the current allocation is updated ($\mathbf{A} \leftarrow \mathbf{A}'$). To escape local optima, transitions with negative rewards (performance degradation) are accepted with probability $\exp(r_k/\psi_k)$, where $\psi_k$ is a temperature parameter that decays according to a cooling schedule (e.g., $\psi_{k+1} = \alpha\psi_k$). This mechanism allows SA to explore broadly in early stages and exploit the optimal region as $\psi_k \rightarrow 0$, ensuring convergence to a high-quality global allocation $\mathbf{A}^*$. The global scheduler optimization loop is demonstrated in Figure 2.

### 4.2.2. LOCAL SCHEDULER

Given a fixed allocation $\mathbf{A}$ from the global scheduler, the local scheduler derives the optimal parallelism configuration $\mathbf{P}$ by performing constrained enumeration (Dechter, 2003) independently for each non-zero allocation element $a_{t,n}$ to determine the corresponding strategy $p_{t,n}$.

**Search space.** For a specific worker type t utilizing $a_{t,n}$ GPUs of type n, the local scheduler searches for the optimal hybrid parallelism strategy defined by the tuple $p_{t,n} = \langle \delta_{DP}, \delta_{TP}, \delta_{EP} \rangle$. The search space $\Phi_{t,n}$ consists of all valid parallelism tuples that satisfy the following constraints:

*C1: Parallelism compatibility.* The sum of tensor parallelism (Shoeybi et al., 2019) degrees across all data parallelism (Li et al., 2023) replicas must equal the total number of allocated GPUs: $\sum_{i=1}^{\delta_{DP}} \delta_{TP}^{(i)} = a_{t,n}, \forall(t, n)$ where $a_{t,n} > 0$.

*C2: Expert placement strategies.* The expert parallelism degree $\delta_{EP}$ must satisfy divisibility constraints with tensor parallelism degree $\delta_{TP}$: (**i**) Intra-replica sharding ($\delta_{EP} \leq \delta_{TP}$) requires $\delta_{EP}$ to divide $\delta_{TP}$ evenly. (**ii**) Cross-replica sharding ($\delta_{EP} > \delta_{TP}$) requires $\delta_{EP}$ to be a multiple of $\delta_{TP}$ and identical tensor parallelism degrees across replicas.

*C3: Topology awareness.* The tensor and expert parallelism degrees must not exceed the number of GPUs per machine $G_{node}$ (e.g., $G_{node} = 8$) to restrict communication-intensive operations to high-bandwidth intra-machine inter-

bottlenecks. The global scheduler iteratively invokes the local scheduler to perform parallelism configuration search and system throughput estimation, as detailed in §4.2.2.

We summarize the MDP formulation (state space, action space, transition dynamics, and reward function) in Appendix C and focus on our key algorithmic contribution.

**Guided generation policy.** Naïve exploration of the MDP's combinatorially large action space wastes iterations on unproductive resource transfers. Our key insight is that resources should flow from high-throughput (over-provisioned) worker types to low-throughput (bottleneck) worker types. We operationalize this with a *guided generation policy* that leverages the throughput profile $\boldsymbol{\Theta} = \{\Theta_{pre}, \Theta_{attn}, \Theta_{ffn}\}$ ($\Theta_t$ represents the effective throughput of worker type t) from the local scheduler (§4.2.2) to bias action selection toward bottleneck resolution:

- The probability of selecting destination worker type $t_{dst}$ is defined by the normalized reciprocal of its throughput (i.e., $\Theta_t^{-1} / \sum_{j \in \mathbf{T}} \Theta_j^{-1}$), ensuring worker types with lower throughput receive resources with higher probability.

- Conversely, source $t_{src}$ is selected proportional to its throughput (i.e., $\Theta_t / \sum_{j \in \mathbf{T}} \Theta_j$), ensuring resources are primarily drawn from worker types with excess capacity.

This guided policy concentrates the search on the most promising region of the action space—transfers that directly address throughput imbalances—while still permitting exploration of alternative allocations.

**Optimization algorithm.** Given the non-convex nature of the optimization landscape, we employ Simulated Annealing (SA) (Chang et al., 2013) to solve the defined MDP efficiently. We initialize the search with a uniform allocation

connects (Narayanan et al., 2021; Rajbhandari et al., 2022; Jiang et al., 2023): $\max(\delta_{\text{TP}}, \delta_{\text{EP}}) \leq G_{\text{node}}$.

**Parallelism strategy optimization.** The local scheduler employs an inference task simulator[2] to estimate the achievable throughput for every candidate parallelism strategy $p_{t,n} \in \Phi_{t,n}$ satisfying constraints *C1*-*C3*. Subsequently, it selects the optimal strategy $p_{t,n}^*$ by maximizing this estimated metric. The collection of these optimal strategies constitutes the final parallelism configuration $\mathbf{P}^*$.

**Throughput profile $\Theta$.** Beyond determining the optimal configuration, the local scheduler must quantify the effective throughput each worker type t achieves to guide the global scheduler. Specifically, it synthesizes $\Theta = \{\Theta_{\text{pre}}, \Theta_{\text{attn}}, \Theta_{\text{ffn}}\}$ (referenced in §4.2.1) by aggregating the optimal throughputs across the heterogeneous device groups allocated to each worker type. This throughput profile then serves as the feedback signal for the global MDP, driving subsequent resource transfers.

# 5. Autoscaling Framework

Building upon the scheduling framework (§3), the autoscaling framework is responsible for *dynamically* adjusting resource provisioning and model deployment in response to volatile workloads. Specifically, given the dynamic nature of input workloads—where the request arrival rate $\lambda$ and the input and output token length distributions ($\mu_{\text{in}}, \mu_{\text{out}}$) fluctuate over time—it is essential to autoscale resources to maintain service quality. An effective autoscaling framework should answer two essential questions: *When* and *how* to scale? We answer these questions by demonstrating our autoscaling metric (§5.1) and method (§5.2).

## 5.1. Autoscaling Metric

The autoscaling metric is used to reflect the real-time service load, serving as a signal for triggering scaling decisions (i.e., *when* to scale?). We characterize different candidates on real-world service processes and identify the one that best describes service load fluctuations.

**Candidate metrics.** We consider four principal categories of metrics for autoscaling evaluation: (**i**) Hardware utilization metrics (i.e., GPU utilization); (**ii**) queue-based metrics (i.e., average queue wait time); (**iii**) latency metrics (i.e., Time-To-First-Token (TTFT) and Time-Between-Tokens (TBT)); and (**iv**) throughput metrics (i.e., Tokens-Per-Second (TPS) for prefill and FFN workers, and KV-operations per second for attention workers). An ideal autoscaling metric should satisfy three key properties: It should (**i**) clearly correlate with actual workload changes

---

[2]The simulator takes the incoming workload profile $\mathbf{W}$, allocation element $a_{t,n}$, and parallelism strategy $p_{t,n}$ as inputs and outputs the estimated throughput. Simulation details are shown in §6 and Appendix A.

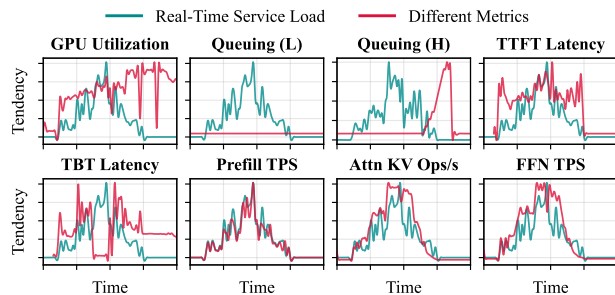

*Figure 3.* Characterization results of different metrics. Queuing (L/H) represents the queuing length under low/high system load.

with a high signal-to-noise ratio, (**ii**) react quickly to load variations without excessive lag, and (**iii**) exhibit a predictable (ideally linear) relationship with resource allocation to provide clear guidance on scaling magnitude.

**Characterization results.** We conducted an empirical analysis of these candidate metrics using a service trace from a production workload (Patel et al., 2024). The trace captures *dynamic workload behaviors*, with time-varying distributions of input/output sequence lengths and arrival rates. To evaluate the effectiveness of each metric, we tested whether its real-time fluctuations aligned with the actual service load (i.e., KV cache-missed prefill input[3]). Figure 3 demonstrates the characterization results for each of the metric categories: (**i**) *Hardware utilization metrics* are misleading, remaining high even under low workload due to memory-bound operations. (**ii**) *Queue-based metrics* are inherently reactive, signaling issues only after queues have accumulated (i.e., under high system load). (**iii**) *Latency metrics* exhibit non-linear, cliff-like behavior, spiking only after SLOs are violated. (**iv**) In contrast, *throughput metrics* correlate strongly with workload changes, enabling proactive and proportional scaling. Based on these observations and analysis, we select throughput metrics as the primary autoscaling metric.

**Autoscaling signal.** Let $\Upsilon_t^{\text{target}}$ denote the target throughput[4] (i.e., TPS for prefill and FFN workers, and KV-operations per second for attention workers) for worker type t, and let $\Upsilon_t^{\text{observed}}$ denote the observed throughput for worker type t. We define the effective throughput pressure as $E_t = \Upsilon_t^{\text{observed}} / \Upsilon_t^{\text{target}}$. A pressure ratio $E_t > 1$ indicates that type-t workers are operating above target capacity and require scaling out, while $E_t < 1$ suggests potential for scaling in. To mitigate instability from transient workload oscillations, we employ a *hysteresis band mechanism* (Lorido-Botran et al., 2014) with distinct thresholds for scaling out ($\varrho_{\text{out}}$,

---

[3]If a request is fully cached, it consumes negligible computational resources. Therefore, the only true load is the work the GPUs must actually perform (i.e., computing new tokens).

[4]The target throughput is determined through pressure testing: Given a service and its workload profile, we incrementally increase load until throughput plateaus despite additional load, then set the target slightly below that point to maintain a safety margin.

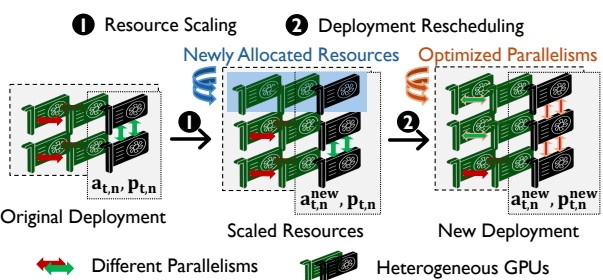

*Figure 4.* Illustration of the HEXGEN-3 autoscaling process.

e.g., 0.1) and scaling in ($\varrho_{in}$, e.g., 0.1). Specifically, a scaling event is triggered only when $E_t > 1 + \varrho_{out}$ or $E_t < 1 - \varrho_{in}$. During online serving, we continuously monitor $E_t$ for each worker type to initiate autoscaling actions independently.

## 5.2. Autoscaling Method

The autoscaling method determines the scaling magnitude and deployment strategy, translating metric changes into concrete resource adjustments (i.e., *how* to scale?). An illustration of the autoscaling process is provided in Figure 4. We demonstrate our autoscaling method as follows.

### 5.2.1. SCALING MECHANISM

Given the throughput pressure $E_t$, the scaling mechanism determines the required scaling magnitude for each worker type $t \in \{pre, attn, ffn\}$ (fine-grained autoscaling).

**Resource scaling strategy.** Once a scaling event is triggered, following established approaches, we employ a *proportional control mechanism* (Lai et al., 2025) designed to bridge the throughput gap efficiently. Unlike step-based scaling, which adds fixed increments, proportional control aims to align system capacity with the real-time service load in a single scaling step. For each specific resource allocation element $a_{t,n}$, the updated element $a_{t,n}^{new} \in \mathbf{A}^{new}$ is computed by projecting the current capacity against the observed pressure $E_t$: $a_{t,n}^{new} = \lceil a_{t,n} \times E_t \rceil, \forall n$. By linearly scaling the resources proportional to the deviation ratio $E_t$, the system rapidly aligns the cluster's aggregate service rate with the incoming traffic intensity.

### 5.2.2. DEPLOYMENT RESCHEDULING

Given the updated workload profile $\mathbf{W}^{new}$ and resource allocation element $a_{t,n}^{new} \in \mathbf{A}^{new}$ derived from the scaling mechanism, the deployment rescheduling module determines the updated parallelism strategy $p_{t,n}^{new} \in \mathbf{P}^{new}$.

**Warm-start optimization.** Solving the parallelism configuration problem from scratch (cold-start) imposes prohibitive latency during real-time autoscaling. To circumvent this, we employ *warm-start optimization* by projecting the currently active strategy $p_{t,n}$ onto the new resource allocation $a_{t,n}^{new}$. Specifically, we preserve the existing tensor- and expert-parallel configurations of $p_{t,n}$ while scaling the data-parallel

replicas to match $a_{t,n}^{new}$. This projection initializes the local scheduler (§4.2.2) with a high-quality candidate, effectively pruning the search space by prioritizing configurations that minimize structural changes.

**Cost-aware parallelism search.** Our local scheduler described in §4.2.2 always selects the parallelism strategy with the highest estimated throughput. However, during online service with resource autoscaling, the reconfiguration cost between strategies (e.g., model reloading) must also be considered during scheduling. Therefore, we integrate the reconfiguration cost into the optimization objective of our local scheduler, enabling *cost-aware* online rescheduling that maximizes the total number of requests processed over a stability window $\mathcal{T}_{window}$[5]. Specifically, a candidate strategy $\hat{p}_{t,n} \in \Phi_{t,n}$ is selected only if the throughput gain outweighs the reconfiguration cost $\tau$[6]:

$$(\Theta(\hat{p}_{t,n}) - \Theta(p_{t,n})) \cdot (\mathcal{T}_{window} - \tau) > \Theta(p_{t,n}) \cdot \tau \quad (1)$$

where $\Theta(\cdot)$ denotes the estimated throughput for a specific parallelism strategy. Formally, the revised optimization objective for our local scheduler is expressed as:

$$p_{t,n}^{new} = \underset{\hat{p}_{t,n} \in \Phi_{t,n}}{\operatorname{argmax}} \left[ \Theta(\hat{p}_{t,n}) \cdot (\mathcal{T}_{window} - \tau) \right] \quad (2)$$

This ensures that expensive operations, such as changing tensor parallelism degrees (which require full model reloading), are triggered only when the performance gain is sufficient to amortize the downtime within the expected window.

## 6. System Implementation

**HEXGEN-3 architecture.** We implement the fully disaggregated architecture of HEXGEN-3 atop SGLang (Zheng et al., 2024), a highly optimized LLM serving framework. We adopt design principles from StepMesh (e.g., GPU-direct RDMA data transfer and ping-pong attention-to-FFN communication) (Wang et al., 2025) to mitigate the communication overhead inherent in AFD serving. These principles effectively mask network latency, ensuring GPU computational resources remain fully saturated.

**Asymmetric AFD parallelism.** Our AFD implementation further supports *asymmetric parallelism*, enabling attention and FFN workers to operate with independent tensor/expert parallelism degrees. To coordinate communication across mismatched parallelism groups, we implement a *rank-0-proxy* pattern, in which only the rank-0 process within each group participates in StepMesh communications, followed by intra-group synchronization via NCCL collectives (NVIDIA Corporation, 2025) (e.g., `Broadcast`).

**Simulation for AFD.** Our simulator models AFD execu-

---
[5]The stability window $\mathcal{T}_{window}$ is derived from the Average Inter-Scaling Interval (AISI) observed in recent historical traces.

[6]The reconfiguration cost $\tau$ is obtained by dividing the model parameter size by the effective memory loading bandwidth; it depends on the specific strategy difference between $\hat{p}_{t,n}$ and $p_{t,n}$.

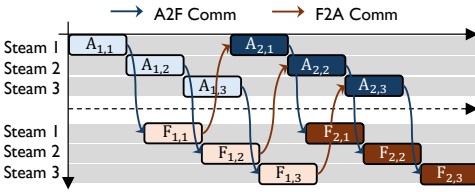

*Figure 5.* Illustration of the AFD dependency simulation. A2F and F2A Comm denote the attention-to-FFN and FFN-to-attention communication operations, respectively. $A_{i,j}$ and $F_{i,j}$ represent the computations performed at the i-th layer for the j-th micro-batch by the attention and FFN workers, respectively.

tion dynamics at *per-layer* and *per-micro-batch* granularity, capturing coordination between attention and FFN workers. It enforces the sequential dependencies of the AFD pipeline—attention computation, attention-to-FFN communication, FFN computation, and FFN-to-attention communication—while accounting for pipelined overlap across layers and micro-batches (see Figure 5). Additionally, attention computation latency is modeled dynamically based on maximum sequence length per micro-batch, accurately reproducing runtime dynamics and resource contention in real-world AFD deployments.

**Request routing and processing details.** In our implementation, a load balancer assigns each request a random `bootstrap_room` and routes it to a prefill worker according to `bootstrap_room % dp_size`. During decoding, workers adopt continuous batching with waiting/running queues and First-Come-First-Served (FCFS) scheduling for newly arrived requests. With AFD enabled, attention workers retain the same continuous batching and FCFS policy, while FFN workers follow the computation schedule of attention workers and process micro-batches as they arrive from the attention side.

Further implementation details (e.g., the soft scaling mechanism during resource changes) are provided in Appendix D.

## 7. Evaluation

In this section, we evaluate HEXGEN-3 under both static resource provisioning and dynamic resource autoscaling.

### 7.1. Experimental Setup

**Hardware environments.** We conduct evaluations in the following two setups: (**i**) *Homogeneous setup*: We rent 16×NVIDIA H100-80G GPUs[7], with a total budget of $49.12/h, to represent the standard homogeneous case; and (**ii**) *heterogeneous setup*: We rent 8×NVIDIA H100-80G, and 16×NVIDIA H20-96G GPUs, with a total budget of $54.16/h, to represent the heterogeneous case.

**Baseline frameworks.** We compare HEXGEN-3 and

---

[7]GPU resource allocation is subject to dynamic adjustment when autoscaling is enabled.

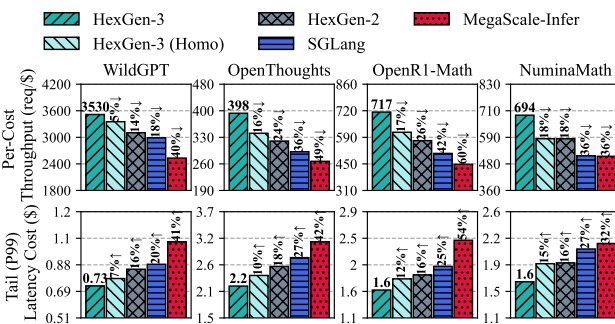

*Figure 6.* End-to-end experimental results comparing HEXGEN-3 with existing LLM serving frameworks. HEXGEN-3 (Homo) represents HEXGEN-3 implemented on the homogeneous setup.

HEXGEN-3 (Autoscale) with the following state-of-the-art LLM serving frameworks: (**i**) SGLang (Zheng et al., 2024), the standard homogeneous baseline with prefill-decoding disaggregation enabled; (**ii**) MegaScale-Infer (Zhu et al., 2025), the homogeneous baseline with AFD enabled; (**iii**) HexGen-2 (Jiang et al., 2025d), the standard heterogeneous baseline with prefill-decoding disaggregation enabled; (**iv**) SGLang enhanced by our autoscaling framework (§5), the standard autoscaling baseline; and (**v**) HeteroScale (Autoscale) (Li et al., 2025), the autoscaling baseline with prefill-decoding disaggregation enabled.

**Models and traces.** We evaluate each baseline framework using Qwen3-30B-A3B model (Bai et al., 2023). For static resource provisioning, we evaluate on four real-world chatbot and reasoning workloads: WildGPT (Zhao et al., 2024a), OpenThoughts (Guha et al., 2025), OpenR1-Math, and NuminaMath (Li et al., 2024); for dynamic resource autoscaling, we evaluate on the trace shown in Table 1. Following prior work (Li et al., 2023), we subsample from each trace and generate inference workloads with Poisson arrivals.

**Evaluation metrics.** Following prior works (Sheng et al., 2023; Yang et al., 2023), we adopt per-dollar throughput (req/$) and P99 latency cost ($) as our primary metrics. These metrics normalize total system throughput and tail latency by the hourly cost of the allocated GPU resources, providing a direct measure of cost-efficiency.

*Table 1.* Trace for autoscaling evaluation. Workload types 1–4 represent WildGPT, OpenThoughts, OpenR1, and NuminaMath.

| Hours | 1 | 2 | 3 | 4 | 5 | 6 |
|---|---|---|---|---|---|---|
| Request Load | 6.6K | 11.2K | 61.4K | 129K | 13.9K | 8.2K |
| Workload Type | 3 | 3 | 1 | 1 | 4 | 2 |

### 7.2. End-to-end Performance

**Static resource provisioning.** We present the end-to-end experimental results comparing HEXGEN-3 with baseline systems under static resource provisioning across different workload traces in Figure 6. Results demonstrate that HEXGEN-3 achieves consistent performance improvements over both homogeneous and heterogeneous baselines.

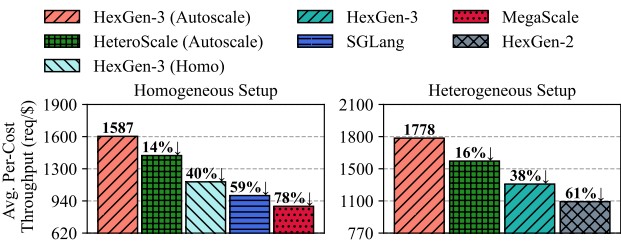

*Figure 7.* End-to-end experimental results comparing HEXGEN-3 with baselines on the highly dynamic workload trace (Table 1).

*Table 2.* Cross-model evaluation on Mixtral-8x22B with 48 H100 GPUs. Throughput is measured in token/s.

| Workload | SGLang | HEXGEN-3 |
|---|---|---|
| NuminaMath CoT | 7924.38 | 9821.40 (23.9%↑) |
| OpenR1-Math | 7765.98 | 8983.50 (15.7%↑) |
| OpenThoughts | 6676.77 | 7622.40 (14.2%↑) |

HEXGEN-3 (Homo) outperforms SGLang by 16.6% on average and up to 21.6%, and outperforms MegaScale-Infer by 28.7% on average and up to 36.8%, demonstrating the effectiveness of our scheduling framework and fine-grained disaggregation. Furthermore, HEXGEN-3 in the heterogeneous configuration outperforms HexGen-2 by 20.3% on average and up to 25.8%, demonstrating that AFD effectively boosts the performance of heterogeneous serving. Finally, HEXGEN-3 in the heterogeneous configuration outperforms the homogeneous configuration by 14.0% on average and up to 17.9%, demonstrating that integrating decoding-optimized heterogeneous GPUs (e.g., H20) can further enhance the cost-efficiency of LLM serving. We also present the TTFT, TBT, inference-time batch sizes, and end-to-end latency of our system in Appendix E.

**Dynamic resource autoscaling.** We present the autoscaling evaluation results under dynamic workloads on the trace in Table 1 in Figure 7. Experimental results demonstrate that HEXGEN-3 (Autoscale) achieves high cost-efficiency by avoiding both resource over-provisioning and performance degradation from under-provisioning. Compared with HeteroScale (Autoscale), HEXGEN-3 (Autoscale)'s finer-grained resource scheduling and autoscaling achieve up to 15.8% and on average 14.8% speedup. Additionally, when compared with other non-autoscaling baselines, HEXGEN-3 (Autoscale) achieves up to 78.3% and on average 55.1% performance improvement. These results collectively demonstrate that autoscaling is essential for maintaining high cost-efficiency in LLM serving.

**Evaluation on larger model and cluster.** We further evaluate HEXGEN-3 on Mixtral-8x22B with 48 H100 GPUs to validate its effectiveness on a larger model and cluster. As shown in Table 2, HEXGEN-3 consistently outperforms SGLang, improving throughput by 23.9% on NuminaMath CoT, 15.7% on OpenR1-Math, and 14.2% on OpenThoughts. These results show that the benefits of AFD and hierarchical scheduling remain effective at larger model

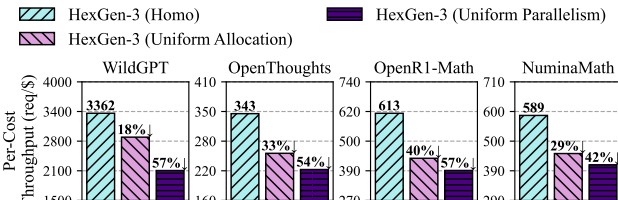

*Figure 8.* Ablation study on allocation and parallelism impact.

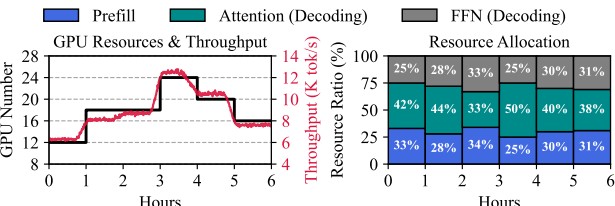

*Figure 9.* Case study on resource allocation for autoscaling.

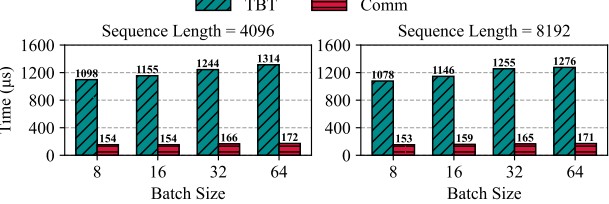

*Figure 10.* Case study on per-layer AFD communication latency.

and cluster scales, confirming that HEXGEN-3 generalizes beyond the Qwen3-30B-A3B setting.

### 7.3. Ablation and Case Study

**Ablation study.** To evaluate our scheduling design, we progressively disable the resource allocation optimization and parallelism strategy optimization, comparing against HEXGEN-3 with uniform configurations in the homogeneous setup in Figure 8. Without resource allocation optimization, the performance of HEXGEN-3 degrades by 22.9% on average and up to 28.6%. If we further disable parallelism strategy optimization, the performance is further reduced by 14.6% on average and up to 25.1%. These results demonstrate that both optimization designs described in §4 are essential in enhancing system performance.

**Case study on allocation and parallelism.** We present a case study of HEXGEN-3 (Autoscale)'s resource allocation and parallelism strategies in Figure 9 and Table 3, both of which adapt dynamically based on the current workload type and request load. For instance, from the first to second hour, the increased request load triggers an increase in resource allocation (12⇒18) to accommodate the higher demand. From the second to third hour, although the total resource allocation remains unchanged, more resources are allocated to the prefill and FFN phases due to workload type changes, and the parallelism strategy adjusts accordingly.

**Case study on AFD communication latency.** We present a case study of the per-layer AFD communication latency

*Table 3.* Case study on parallelism strategies for heterogeneous autoscaling. (2,1) represents 2 replicas with $\delta_{TP}$ equal to 2 and 1.

| Type | 0–1h | 1–2h | 2–3h | 3–4h | 4–5h | 5–6h |
|---|---|---|---|---|---|---|
| **Prefill Workers** | $(2)^H$ | $(2)^H$ | $(2)^H$ | $(2)^H$ | $(2)^H$ | $(2)^H$ |
| | $(2)^L$ | $(2,1)^L$ | $(2,2)^L$ | $(2,2)^L$ | $(2,2)^L$ | $(2,1)^L$ |
| **Attention Workers** | $(1)^H$ | $(1,1)^H$ | $(1,1)^H$ | $(2,2)^H$ | $(1,1)^H$ | $(1,1)^H$ |
| | $(2,2)^L$ | $(2,2,2)^L$ | $(1,1,1,1)^L$ | $(2,2,2,2)^L$ | $(2,2,1,1)^L$ | $(1,1,2)^L$ |
| **FFN Workers** | $(1)^H$ | $(1,1)^H$ | $(1,1)^H$ | $(1,1)^H$ | $(1,1)^H$ | $(1,1)^H$ |
| | $(1,1)^L$ | $(1,1,1)^L$ | $(1,1,1,1)^L$ | $(1,1,1,1)^L$ | $(1,1,1,1)^L$ | $(1,1,1)^L$ |

$^H$ = H100, $^L$ = H20

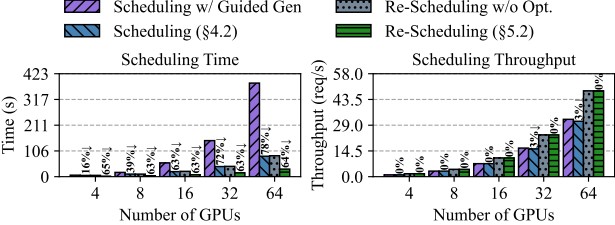

*Figure 11.* Case study on scheduling time and optimality. The rescheduling is tested on cases where the resources extend to $1.5\times$.

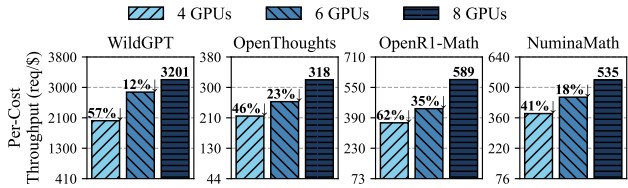

*Figure 12.* Case study on the scalability of HEXGEN-3.

*Table 4.* SLO-constrained throughput comparison on $8\times$H100 GPUs under a 50 ms TPOT SLO. Throughput is measured in token/s, and ITL is measured in ms.

| System | Workload | Throughput | Mean | P50 | P90 | P95 |
|---|---|---|---|---|---|---|
| SGLang | NuminaMath | 6760 | 32.88 | 32.47 | 32.99 | 33.33 |
| HEXGEN-3 | NuminaMath | 7731 (14.4%↑) | 47.92 | 47.01 | 48.28 | 48.73 |
| SGLang | OpenR1-Math | 6554 | 33.26 | 33.20 | 34.05 | 34.43 |
| HEXGEN-3 | OpenR1-Math | 7632 (16.5%↑) | 46.63 | 46.44 | 47.21 | 47.87 |
| SGLang | OpenThoughts | 4995 | 35.56 | 34.69 | 39.95 | 42.12 |
| HEXGEN-3 | OpenThoughts | 5992 (20.0%↑) | 45.17 | 44.89 | 45.93 | 46.34 |
| SGLang | WildGPT | 15154 | 34.22 | 34.05 | 35.69 | 36.52 |
| HEXGEN-3 | WildGPT | 16687 (10.1%↑) | 48.34 | 48.05 | 49.32 | 49.41 |

across varying batch sizes and sequence lengths in Figure 10. The results demonstrate that the communication latency accounts for only 13.6% on average and up to 14.2% of the overall per-token generation latency (i.e., TBT), validating the efficiency of our AFD implementation. Furthermore, during token generation, this communication latency is further minimized by overlapping with computation (detailed in §6), enabling efficient AFD serving with minimal additional communication overhead.

**Case study on scheduling efficiency and optimality.** We evaluate the efficiency and optimality of our scheduling algorithms in Figure 11. For efficiency evaluation, we compare against two baselines: (**i**) Scheduling w/o Guided Gen, which replaces our guided generation heuristic in §4.2 with exhaustive search over all possible configurations, and (**ii**) Re-Scheduling w/o Opt., which disables the warm-start and cost-aware optimizations described in §5.2.2. Results demonstrate that our scheduling algorithm achieves up to 78.2% (on average 53.7%) reduction in scheduling time compared to exhaustive search, while our rescheduling algorithm achieves up to 65.5% (on average 63.6%) reduction compared to the unoptimized baseline. For optimality evaluation, we compare the throughput achieved by our heuristic-based algorithms against their respective baselines. Results show that our scheduling algorithm achieves comparable throughput (within 3.4% difference) compared to exhaustive search, and our rescheduling algorithm maintains identical throughput to the unoptimized baseline. These results show that our optimizations significantly reduce computational overhead while maintaining near-optimal solution quality.

**Case study on scalability.** We present the cost-efficiency results of HEXGEN-3 with increasing GPU allocation in Figure 12. Results show that as the number of GPUs increases, the per-cost throughput of HEXGEN-3 improves by 23.3% on average and up to 37.1% across different workload traces. This is because more GPUs offer greater flexibil-

ity in resource allocation and parallelism optimization for our *fully* disaggregated architecture, thus achieving better cost-efficiency with increased GPU capacity.

**Case study on SLO-constrained evaluation.** We further conduct an SLO-constrained evaluation on an $8\times$H100 GPU cluster, comparing HEXGEN-3 with SGLang under a common 50 ms TPOT SLO. For a controlled comparison, we also optimize the parallelism strategy of SGLang under the same SLO constraint, and report the best feasible throughput it can achieve. Under this setting, the scheduling objective of HEXGEN-3 is adjusted to maximize throughput among feasible configurations whose TPOT satisfies the SLO constraint. As shown in Table 4, HEXGEN-3 achieves 10.1%–20.0% higher throughput than SGLang across all four workloads, while maintaining P95 ITL below 50 ms in all cases. Notably, HEXGEN-3 allocates more resources to FFN workers than attention workers, e.g., 4 GPUs for FFN versus 2 GPUs for attention, to keep TPOT within the SLO, since FFN computation is the dominant contributor to per-token latency in the disaggregated decoding pipeline. These results demonstrate that HEXGEN-3 preserves its throughput advantage even under a controlled latency constraint, validating its effectiveness for SLO-aware LLM serving.

## 8. Conclusion

We presented HEXGEN-3, a cost-effective LLM serving framework that combines fully disaggregated inference with fine-grained heterogeneous resource autoscaling. HEXGEN-3 introduces a hierarchical scheduling framework for joint resource allocation and parallelism optimization, alongside an autoscaling framework that dynamically adjusts deployments based on throughput metrics. Evaluations demonstrate that HEXGEN-3 achieves up to 60% (on average 46.5%) improvement in per-cost throughput under static resource provisioning and up to 78.3% (on average 55.1%) improvement with autoscaling under dynamic workloads.

## Acknowledgements

This work is supported by the HKUST startup grant R9895 from CSE; RGC-ECS project 26218024; RGC-NSFC project CRS HKUST601/24; Huawei-HKUST joint lab project 24250360Y043.

## Impact Statement

This paper presents work whose goal is to advance the field of Machine Learning. There are many potential societal consequences of our work, none which we feel must be specifically highlighted here.

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

# A. Simulator Details

We implement a lightweight simulator for LLM inference serving that estimates request-level latency and worker-type throughput under heterogeneous GPU clusters and disaggregated serving architectures. The simulator maintains a global simulation clock and processes a priority queue of events (e.g., *arrival*, *prefill*, *decode*, *completion*) in timestamp order. This discrete-event structure captures queueing and resource contention effects while remaining lightweight enough to serve as the inner loop for enumerating candidate parallelism strategies in §4.2.2.

## A.1. Inputs and Outputs

**Workload profile.** The simulator takes the *workload profile* $\mathbf{W}$ as input:

$$\mathbf{W} \triangleq \langle \lambda, \ \mu_{\text{in}}, \ \mu_{\text{out}} \rangle,$$

where $\lambda$ is the request arrival rate, and $\mu_{\text{in}}$ / $\mu_{\text{out}}$ are the (empirical) distributions of input/output token lengths derived from a sliding window of recent traffic. Unless otherwise specified, arrivals are generated by a Poisson process with rate $\lambda$, and each request's input/output token lengths are sampled from $\mu_{\text{in}}$ and $\mu_{\text{out}}$, respectively.

**Cluster provisioning and deployment strategy.** The simulator takes the heterogeneous cluster capacity $\mathbf{D} = \{d_n\}_{n=1}^{N}$ and a candidate deployment strategy $\langle \mathbf{A}, \mathbf{P} \rangle$ as inputs, where $\mathbf{A} \in \mathbb{Z}_{\geq 0}^{3 \times N}$ is the resource allocation matrix and $\mathbf{P} = \{\mathbf{P}_{\text{pre}}, \mathbf{P}_{\text{attn}}, \mathbf{P}_{\text{ffn}}\}$ is the parallelism configuration, as defined in §4.1. Each non-zero allocation element $a_{t,n}$ (worker type $t \in \{\text{pre}, \text{attn}, \text{ffn}\}$ on GPU type $n$) is paired with a parallelism strategy $p_{t,n} = \langle \delta_{\text{DP}}, \delta_{\text{TP}}, \delta_{\text{EP}} \rangle$. Network bandwidth parameters are also provided for (i) KV-cache transfer and (ii) activation transfer, consistent with the disaggregated pipeline in Figure 1.

**Scheduling / routing policy.** The simulator supports multiple request-to-worker routing policies (e.g., round-robin, load-aware, and roofline-informed heuristics). This routing determines which worker pool processes each request and therefore shapes queueing delay via load balance.

**Outputs.** For each request, the simulator records a completion time $c_i$ and latency $L_i = c_i - t_i$, and reports aggregate tail metrics (e.g., $\text{P50}(L)$, $\text{P95}(L)$, $\text{P99}(L)$). Most importantly for the schedulers, it estimates the effective throughput of each worker type,

$$\mathbf{\Theta} = \{\Theta_{\text{pre}}, \Theta_{\text{attn}}, \Theta_{\text{ffn}}\},$$

## A.2. Performance Model

**Roofline-inspired kernel timing.** We use a lightweight roofline-style model to estimate GPU execution time for each phase/operation. For a kernel class with arithmetic intensity AI, the effective performance is bounded by compute and memory bandwidth:

$$\text{Perf} = \min\left(\text{PeakFLOPs}, \ \text{BW} \cdot \text{AI}\right).$$

Given model architecture parameters (e.g., hidden size, number of layers) and request shape (sequence length and batch size), the model returns time estimates for prefill and decoding, with decoding further decomposed into attention and FFN components to match the fully disaggregated setting.

**Prefill and decoding decomposition.** Under workload profile $\mathbf{W}$, each request experiences (i) queueing delay, (ii) prefill time, and (iii) decoding time:

$$T \approx T^{\text{queue}} + T^{\text{pre}}(\ell^{\text{in}}) + T^{\text{dec}}(\ell^{\text{out}}),$$

where $\ell^{\text{in}} \sim \mu_{\text{in}}$ and $\ell^{\text{out}} \sim \mu_{\text{out}}$. Continuous batching is applied, bounded by per-worker batch-size constraints.

**Communication in disaggregation.** Disaggregated variants incur transfer latencies modeled as size-over-bandwidth. For KV-cache transfer (e.g., Prefill → Attention), for a request with context length $\ell$:

$$T_{\text{KV}}(\ell) = \frac{S_{\text{KV}}(\ell)}{\beta_{\text{KV}}},$$

where $S_{\text{KV}}(\ell)$ is the KV-cache size (bytes) and $\beta_{\text{KV}}$ is the effective KV-transfer bandwidth (bytes/s). For activation transfer between Attention and FFN workers, for a micro-batch of size $b$:

$$T_{\text{act}}(b) = \frac{S_{\text{act}}(b)}{\beta_{\text{act}}},$$

where $S_{\text{act}}(b)$ scales with hidden size and $b$ (FP16 activations by default). For AFD, the simulator respects the per-layer, per-micro-batch dependency order (Attention compute → A2F comm → FFN compute → F2A comm) and supports overlapped pipelining when enabled.

## A.3. Assumptions and Limitations

The simulator is intended for *comparative* evaluation across allocations and parallelism strategies rather than cycle-accurate prediction. Key assumptions include: (i) deterministic hardware performance (no stochastic variance/thermal effects), (ii) fixed model architecture within each run, (iii) network contention is approximated by bandwidth-based transfer delay (no explicit multi-flow contention model), and (iv) failures are ignored unless explicitly enabled. Despite these simplifications, the discrete-event structure preserves the dominant effects that drive serving performance: queueing under non-uniform arrivals, capacity limits from continuous batching, and contention across replicated worker pools, while producing the worker-type throughput profile $\mathbf{\Theta}$ required by §4.2.2.

# B. Extended Background and Related Work

**Disaggregated LLM serving.** Since different inference phases/operations exhibit significantly different computational characteristics, recent efforts (Zhong et al., 2024; Patel et al., 2024; Chen et al., 2024) propose utilizing separate hardware resources for each phase/operation to avoid performance interference. In a fully disaggregated LLM serving architecture, the system deploys three types of specialized model replicas (workers): Prefill, attention, and FFN workers, with inter-worker communication to exchange KV cache and intermediate activations. This architecture enables each worker type to adopt tailored resource allocations and parallelism configurations based on its specific computational requirements, achieving better resource utilization and higher serving throughput compared to monolithic architectures. However, existing phase splitting frameworks either (**i**) fail to *fully* disaggregate inference phases with distinct computational characteristics (Hu et al., 2024; Li et al., 2025) or (**ii**) enable deployment scheduling based on *static* resource provisioning (Qin et al., 2024; Zhu et al., 2025; Jiang et al., 2025b), leading to inefficient resource utilization under dynamic workloads.

**Heterogeneous LLM serving.** Heterogeneous LLM serving refers to systems that utilize a mix of GPU types with varying hardware characteristics—such as different computational and memory capacities—to optimize resource utilization and cost-efficiency (Miao et al., 2024; Mao et al., 2025; Jiang et al., 2025a;b;d; Peng et al., 2026; Liang et al., 2026; Tong et al., 2025; Jiang et al., 2026b; Peng et al., 2025). HexGen (Jiang et al., 2023) proposes asymmetric partitioning and advanced scheduling techniques to deploy generative inference in decentralized and heterogeneous settings. Helix (Mei et al., 2025) formulates the problem of allocating heterogeneous GPUs and network connections as a maximum flow problem and adopts a mixed integer linear programming algorithm to discover optimized strategies for serving LLMs. These works demonstrate that strategic utilization of heterogeneous resources significantly enhances the cost-efficiency of LLM serving. HEXGEN-3 advances this paradigm by optimizing heterogeneous model deployment within a *fully* disaggregated architecture that separates prefill, attention, and FFN workers, thereby maximizing the utilization of heterogeneous computational resources.

**Autoscaling in LLM serving.** Autoscaling dynamically adjusts computational resource allocation in response to fluctuating workload demands, which is essential for maintaining quality of service while minimizing operational costs in LLM serving (Fu et al., 2024; Sun et al., 2024). Recent advances have addressed key challenges in this domain. HeteroScale (Li et al., 2025) tackles coordinated autoscaling through the first large-scale empirical analysis of autoscaling metrics in production environments, providing insights into

effective resource management strategies. TokenScale (Lai et al., 2025) introduces a token-velocity metric that enables independent scaling of model workers based on their specific computational bottlenecks. These approaches demonstrate that autoscaling mechanisms can effectively reduce operational costs in LLM serving systems under dynamic workload conditions. Distinctively, HEXGEN-3 employs *fine-grained* autoscaling that decouples the scaling of attention and FFN workers, thereby effectively mitigating resource stranding arising from workload heterogeneity.

**Hybrid model parallelism.** Hybrid model parallelism combines tensor parallelism with pipeline parallelism to efficiently scale LLM training and inference beyond what either strategy achieves alone (Zheng et al., 2022; Li et al., 2023; Miao et al., 2022; Jiang et al., 2022; Wang et al., 2024; He et al., 2026; Yan et al., 2025). Hybrid parallelism enables fitting massive models across multiple GPUs while minimizing inter-machine communication overhead and latency, which is critical for meeting real-time inference requirements.

# C. Global Scheduler Details

We utilize a global scheduler to search for the resource allocation matrix $\mathbf{A}$, formulating this search process as a finite-horizon Markov Decision Process (MDP) (Puterman, 2014). In this framework, the global scheduler acts as an intelligent agent, sequentially refining the allocation by transferring resources between worker types to resolve bottlenecks. The global scheduler iteratively invokes the local scheduler to perform parallelism configuration search and system throughput estimation, as detailed in §4.2.2.

**State space.** The state $s_k$ at search step k captures the current resource allocation and its resulting system throughput: $s_k = [\mathbf{A}, \mathbf{\Theta}]$, where $\mathbf{A}$ is the current resource allocation matrix, and $\mathbf{\Theta} = \{\Theta_{pre}, \Theta_{attn}, \Theta_{ffn}\}$ is the throughput profile where $\Theta_{pre}$, $\Theta_{attn}$, and $\Theta_{ffn}$ represent the effective throughput of the prefill, attention, and FFN workers, respectively. For each $\mathbf{A}$, the corresponding $\mathbf{\Theta}$ is obtained by invoking the local scheduler (§4.2.2).

**Action space.** We define the action $u_k$ available to the global scheduler as a resource transfer tuple: $u_k = \langle t_{src}, t_{dst}, n \rangle$, where $t_{src} \in \mathbf{T} \cup \{\emptyset\}$ represents the source worker type, $t_{dst} \in \mathbf{T}$ denotes the destination worker type, and $n \in \{1, \ldots, N\}$ specifies the target GPU type. Executing this action moves a fixed block of b GPUs[8] from $t_{src}$ to $t_{dst}$.

**Transition dynamics.** The transition to the next state $s_{k+1}$ is governed by a deterministic resource transfer. When the global scheduler executes action $u_k$, it updates the allocation

---

[8]The block size is determined by the model size (typically $\{1, 2, 4, 8\}$); e.g., 4 GPUs for 70B models.

as: $a_{t_{dst},n} \leftarrow a_{t_{dst},n} + b$, $a_{t_{src},n} \leftarrow a_{t_{src},n} - b$. This transition is valid only if $a_{t_{src},n} \geq b$. Following the update, the throughput profile $\Theta$ is refreshed by invoking the local scheduler.

**Guided generation policy.** To accelerate convergence, we employ a *guided generation policy* that enables the global scheduler to leverage the throughput profile $\Theta$ obtained from the local scheduler. This policy prioritizes resource transfers from high-throughput to low-throughput worker types: The probability of selecting a destination worker type $t_{dst}$ is defined by the normalized reciprocal of its throughput (i.e., $\Theta_t^{-1}/\sum_{j\in\mathbf{T}}\Theta_j^{-1}$), while the source $t_{src}$ is selected proportional to its throughput (i.e., $\Theta_t/\sum_{j\in\mathbf{T}}\Theta_j$). This biases action generation toward resolving bottlenecks.

**Reward function.** The reward signal guides the global scheduler toward the global optimum by quantifying the performance gain of each action. We define the reward $r_k$ as the bottleneck throughput improvement: $r_k = \min(\Theta(s_{k+1})) - \min(\Theta(s_k))$. Positive values reinforce actions that increase the throughput of the slowest worker type, providing a directional signal for optimization.

**Optimization algorithm.** Given the non-convex nature of the optimization landscape, we employ Simulated Annealing (SA) (Chang et al., 2013) to solve the defined MDP efficiently. We initialize the search with a uniform allocation $\mathbf{A}_0$, where available GPUs are distributed evenly among worker types. At each iteration k, SA generates a candidate action $u_k$ according to the guided generation policy to transition from state $s_k$ to $s_{k+1}$, thereby generating a new allocation $\mathbf{A}'$. The transition is accepted unconditionally if it yields a positive reward ($r_k > 0$, indicating throughput improvement) and the current allocation is updated ($\mathbf{A} \leftarrow \mathbf{A}'$). To escape local optima, transitions with negative rewards (performance degradation) are accepted with probability $\exp(r_k/\psi_k)$, where $\psi_k$ is a temperature parameter that decays according to a cooling schedule (e.g., $\psi_{k+1} = \alpha\psi_k$). This mechanism allows SA to explore broadly in early stages and exploit the optimal region as $\psi_k \to 0$, ensuring convergence to a high-quality global allocation $\mathbf{A}^*$. The global scheduler optimization loop is demonstrated in Figure 2.

## D. System Implementation Details

**AFD architecture.** We implement the AFD architecture of HEXGEN-3 atop SGLang (Zheng et al., 2024), a highly optimized LLM serving framework. To mitigate the communication overhead inherent in AFD serving, we adopt two high-performance design principles from StepMesh (Wang et al., 2025): (**i**) We leverage GPUDirect RDMA to enable zero-copy data transfers between attention and FFN workers; and (**ii**) we employ a ping-pong pipelining strategy that overlaps communication with computation by transmitting activation tensors for the subsequent micro-batch while

the worker processes the current micro-batch. Collectively, these mechanisms effectively mask the network latency inherent to AFD, ensuring that GPU computational resources remain fully saturated.

**Asymmetric AFD parallelism.** Our AFD implementation further supports *asymmetric parallelism*, enabling attention and FFN workers to operate with independent tensor/expert parallelism degrees. To coordinate communication across mismatched parallelism groups, we implement a *rank-0-proxy* pattern, in which only the rank-0 process within each group participates in StepMesh communications, followed by intra-group synchronization via NCCL collectives (NVIDIA Corporation, 2025) (e.g., `Broadcast`). This design enables both worker types to be configured and scaled independently within the AFD architecture.

**Simulation for AFD.** Our simulator models the execution dynamics of the AFD architecture by capturing the coordination between attention and FFN workers at *per-layer and per-micro-batch granularity*. The simulation enforces the strict sequential dependencies of the AFD generation pipeline—spanning attention computation, attention-to-FFN communication, FFN computation, and FFN-to-attention communication—while accounting for the pipelined overlap of computation across layers and micro-batches, as illustrated in Figure 5. Additionally, attention computation latency is modeled dynamically based on the maximum sequence length within each micro-batch. This granular modeling approach ensures that the simulation accurately reproduces the complex runtime dynamics and resource contention observed in real-world AFD deployments.

**Soft scaling-in mechanism.** To ensure service continuity during scale-in operations, we implement a *soft scaling-in mechanism*. Upon a scale-in decision, target resources are withdrawn from service discovery rather than terminated immediately, allowing active requests to complete while preventing new assignments. During this draining period, our autoscaling framework continuously monitors the throughput of the remaining active workers. If a scale-out signal is triggered—indicating that the reduced capacity is insufficient to sustain the workload—draining resources are immediately reinstated to absorb excess load, circumventing the prohibitive latency of cold-starting new replicas. Termination occurs only after: (**i**) All active requests on target resources have completed, and (**ii**) the remaining active workers demonstrates sustained stability.

## E. TTFT and TBT Results

**TTFT, TBT, batch size, and end-to-end latency results.** We present the TTFT, TBT, inference-time batch size, and end-to-end latency results comparing HEXGEN-3 with baselines in Figure 13. With both prefill-decoding disaggre-

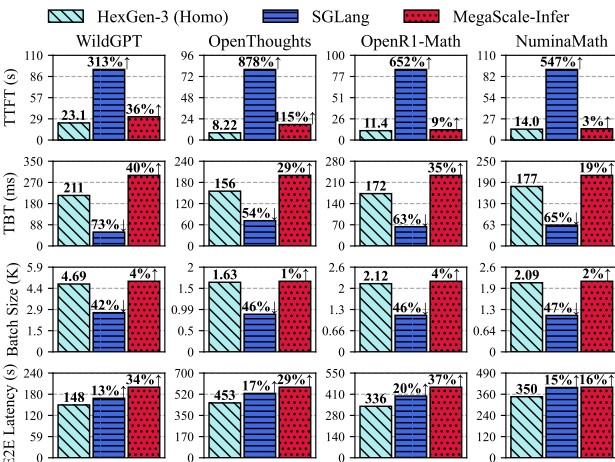

*Figure 13.* TTFT, TBT, inference-time batch size, and end-to-end latency results comparing HEXGEN-3 with SGLang and MegaScale-Infer in the homogeneous setup.

gation and AFD enabled, HEXGEN-3 achieves an 84.2% average reduction in TTFT compared to SGLang and 22.8% compared to MegaScale-Infer. However, HEXGEN-3 and MegaScale-Infer experience a 64% and 87% average increase in TBT, respectively. This TBT increase is due to the AFD architecture enabling a 2.2× average increase in inference-time batch size compared to SGLang—a tradeoff consistent with prior observations on AFD-based systems (Zhu et al., 2025; Chen et al., 2024). Despite the increased TBT, HEXGEN-3 achieves a 16.3% average reduction in end-to-end latency compared to SGLang and 29% compared to MegaScale-Infer, validating its effectiveness for interactive online LLM services in most cases.

