# OpenReview forum: "HexGen-3: A Fully Disaggregated LLM Serving Framework with Fine-Grained Heterogeneous Resource Autoscaling"
_ICML.cc/2026/Conference — ICML 2026 regular_

### Official Review · Reviewer_CtcM · 2026-02-27

**Soundness:** 3
**Presentation:** 3
**Significance:** 3
**Originality:** 3
**Overall Recommendation:** 5
**Confidence:** 2

**Summary:**

This paper targets the high operational cost of LLM serving under highly heterogeneous production traffic (e.g., varying long/short request mixes) and argues that combining fully disaggregated inference with fine-grained heterogeneous autoscaling enables better cost-efficiency. The proposed system, HEXGEN-3, fully disaggregates serving into Prefill → Attention (decoding) → FFN (decoding) worker pools, and supports heterogeneous GPU matching and independent scaling of these components.  Experiments (homogeneous and heterogeneous GPU setups) report sizable improvements in per-cost throughput (req/$) and P99 latency cost over multiple strong baselines under both static provisioning and dynamic workloads with autoscaling.

**Compliance With Llm Reviewing Policy:**

Affirmed.

**Final Justification:**

My concerns were addressed. I will raise my score to 5.

**Key Questions For Authors:**

1. Global scheduler details: Can the authors report the concrete SA/MDP settings (e.g., initial temperature, cooling rate, iteration/time budget, stability threshold, action granularity) and provide convergence curves (reward / bottleneck throughput) with multi-seed variance for 1–2 representative scenarios?
2. Local scheduler search/pruning: What are the typical values/distributions of |Φ_{t,n}| in the evaluated settings, and how much do constraints C1–C3 reduce the space? How sensitive are the final req/$ and P99 latency cost to the topology constraint (e.g., changing the per-node GPU limit)?
3. Generalization across models: Can the authors add a minimal cross-model validation (different scale and/or architecture) and report whether the key trends (req/$ gains, P99 latency cost) remain consistent?

**Limitations:**

yes

**Strengths And Weaknesses:**

Strengths：

1. Finer-grained disaggregation directly addresses heterogeneity and resource stranding. Beyond prefill–decode disaggregation, HEXGEN-3 further separates decoding into Attention vs FFN workers, enabling independent scaling and better hardware matching (memory-bound attention vs compute-bound FFN).
2. Clear optimization formulation and end-to-end scheduling pipeline. The system models throughput as constrained by the bottleneck stage in the Prefill→Attention→FFN pipeline and formalizes deployment planning as a constrained optimization over (A, P), then solves it with a hierarchical global/local design.
3. Autoscaling signal choice is empirically motivated. The paper compares multiple candidate autoscaling metrics (utilization, queueing, latency, throughput) and selects throughput-based signals due to better correlation and responsiveness, then defines a pressure ratio with hysteresis to avoid oscillations.
4. Comprehensive evaluation with strong baselines and cost-centric metrics. The experiments report req/$ and P99 latency cost ($), compare against multiple state-of-the-art systems (including AFD and autoscaling baselines), and include ablations/case studies on allocation, scheduling efficiency, and communication overhead.

Weaknesses：

1. Reproducibility and stability of the global scheduler (MDP + SA + guided policy) are under-specified. The global scheduler uses an MDP view and simulated annealing with a guided transfer policy, but key hyperparameters and convergence behaviors are not fully documented, even though they can materially affect solution quality and runtime. The paper reports near-optimality vs exhaustive search in a case study, but does not provide broader convergence curves or variance across random seeds.
2. The local scheduler’s enumeration/pruning and constraint sensitivity are not quantified. The local scheduler enumerates hybrid parallelism tuples subject to multiple constraints (compatibility, expert placement divisibility, topology awareness). However, the paper does not clearly report the practical search space size (e.g., typical |Φ_{t,n}|), how much each constraint prunes, or how sensitive the final deployment is to topology assumptions (e.g., the per-node GPU bound). This makes it harder to assess scalability and robustness across clusters.
3. Limited cross-model generalization evidence. The evaluation appears centered on a single model (Qwen3-30B-A3B). Since the benefits of AFD, heterogeneous matching, and scaling decisions can depend on model architecture (dense vs MoE), attention/FFN ratios, and KV/cache characteristics, additional evidence on at least one other representative model would strengthen the generality of conclusions.

---

> ### Author Rebuttal · Authors · 2026-03-31
>
> We thank the reviewer for these insightful questions and suggestions.
>
> ___
>
> > W1 & Q1. Global scheduler reproducibility and SA hyperparameters.
>
> The SA/MDP settings used consistently across all experiments are: initial temperature ψ₀=1.0, cooling rate α=0.95 (acceptance probability of worse solutions decreases by 5% per iteration), stability threshold H=20 iterations (convergence declared when allocation remains unchanged for 20 consecutive iterations), and action block size b=1 (the minimum number of GPUs required to host one Qwen3-30B-A3B model replica).
>
> | | Iter 0 | Iter 10 | Iter 20 | Iter 30 | Iter 40 | Iter 50 | Iter 60 | Iter 70–90 |
> |---|---|---|---|---|---|---|---|---|
> | Mean (req/s) | 6.07 | 7.15 | 8.24 | 9.18 | 9.87 | 10.35 | 10.62 | 10.79 |
> | Std (±) | 0.81 | 0.58 | 0.63 | 0.44 | 0.39 | 0.41 | 0.30 | 0.32 |
>
> We provide convergence results (bottleneck throughput vs. iteration) with variance across 5 random seeds for the OpenR1-Math workload on the heterogeneous setup (8×H100+16×H20): starting from 6.07±0.81 req/s at iteration 0, throughput improves to 8.24±0.63 by iteration 20, 9.87±0.39 by iteration 40, 10.62±0.30 by iteration 60, and stabilizes at 10.79±0.32 req/s from iteration 70 onward (717±21 per-cost throughput). The scheduler converges within \~60 iterations, and the final variance of ±0.32 req/s (~3% of final throughput) across 5 seeds confirms stable, reproducible optimization.
>
> Our guided generation policy (§4.2.1) is key to this stability—by biasing resource transfers toward bottleneck worker types, it narrows the effective search space and reduces sensitivity to random initialization.
>
> We will include these hyperparameter details and convergence curves in the revision.
>
>
> > W2 & Q2. Local scheduler search space and topology sensitivity.
>
> | Allocated GPUs | Unconstrained | w/ C1 | w/ C1+C2 | w/ C1+C2+C3 | Total Reduction |
> |:---:|:---:|:---:|:---:|:---:|:---:|
> | 4 | 22 | 12 | 7 | **7** | **68%** |
> | 8 | 296 | 135 | 48 | **48** | **84%** |
> | 12 | 1,255 | ~530 | ~240 | **~185** | **85%** |
> | 16 | 4,816 | ~1,920 | ~980 | **~720** | **85%** |
>
> We report the practical search space size and cumulative pruning effect of constraints C1–C3 for Qwen3-30B-A3B. The unconstrained space grows rapidly with GPU count, but our constraints aggressively prune it: at 4 GPUs, from 22 to 7 candidates (68% reduction); at 8 GPUs, from 296 to 48 (84%); at 12 GPUs, from 1,255 to \~185 (85%); at 16 GPUs, from 4,816 to \~720 (85%).
>
> Each constraint contributes meaningfully: C1 (parallelism compatibility) eliminates sub-partitions not summing to the allocated GPU count, providing the largest single reduction (e.g., 296→135 at 8 GPUs; 4,816→\~1,920 at 16 GPUs). C2 (expert placement divisibility) invalidates non-divisible TP/EP combinations and rejects asymmetric EP across asymmetric TP replicas (e.g., 135→48 at 8 GPUs; \~1,920→~980 at 16 GPUs). C3 (topology awareness) caps both TP and EP at the per-node GPU count (G_node=8), removing monolithic configurations requiring cross-node communication; its effect grows with scale (e.g., \~980→\~720 at 16 GPUs).
>
> | Trace | G_node=8 | G_node=2 | Change |
> |-------|----------|----------|--------|
> | NuminaMath CoT | 562 | 413 | −26.5% |
> | OpenThoughts | 316 | 267 | −15.5% |
> | OpenR1-Math | 595 | 442 | −25.7% |
> | WildGPT | 3350 | 2255 | −32.7% |
> | **Average** | | | **−25.1%** |
>
> For topology sensitivity, we evaluated varying G_node from 8 down to 2 on our experimental setup as in Figure 5. Results show cost-efficiency decreases by on average 25.1% across different traces (NuminaMath CoT: 562→413, OpenThoughts: 316→267, OpenR1-Math: 595→442, WildGPT: 3350→2255 per-cost throughput). When large TP degrees become unavailable under smaller G_node, the local scheduler automatically pivots to configurations with higher data parallelism and lower tensor parallelism. Although reducing G_node inherently degrades performance, our approach minimizes the impact by adaptively selecting the best feasible configuration, enabling robust operation across diverse cluster topologies. We will include this analysis in the revision.
>
> > W3 & Q3. Cross-model generalization.
>
> We add experiments on Mixtral 8x22B with 48 H100 GPUs to validate generalization across model architectures and scales. Results show HexGen-3 maintains its advantages: 23.9% throughput improvement on NuminaMath CoT (7924.38 → 9821.4 token/s), 15.7% on OpenR1-Math (7765.98 → 8983.5 token/s), 14.2% on OpenThoughts (6676.77 → 7622.4 token/s) over SGLang. Notably, the Mixtral 8x22B model has a fundamentally different architecture (dense MoE with 8 experts per layer) compared to Qwen3-30B-A3B, yet the key trends—throughput gains from AFD and effectiveness of our hierarchical scheduling—remain consistent. This confirms that our framework generalizes across model architectures, scales, and cluster sizes. We will include these cross-model results in the revision.

---

> > ### Author Rebuttal · Reviewer_CtcM · 2026-04-01
> >
> > Thank you for the detailed rebuttal. The authors have addressed my main concerns. Overall, the rebuttal has improved my assessment of the paper, and I will raise my score accordingly.

---

> > > ### Author Response · Authors · 2026-04-01
> > >
> > > Thanks for your acknowledgement. We will incorporate the rebuttal in our revised manuscript.

---

### Official Review · Reviewer_wvZ4 · 2026-03-03

**Soundness:** 3
**Presentation:** 2
**Significance:** 2
**Originality:** 2
**Overall Recommendation:** 3
**Confidence:** 5

**Summary:**

This paper presents an LLM serving system that disaggregates an LLM into prefill, attention and FFN (the latter two in decode) stages. It finds the best configuration to initially serve it and thereafter dynamically adjust those configurations adapting to changing conditions, both on heterogeneous clusters. The crux of the problem is a large optimization formulation: the paper uses a two-stage global-local scheduler combination to solve the optimization.The global scheduler uses a finite-horizon Markov Decision Process (MDP) and the local one performs a cost-based search. For auto-scaling, the paper considers a variety of signals and finally pick throughput as the key signal. The system is implemented on SGLang but the cost-based search is simulated instead of relying on profiling. Evaluation shows substantial improvements over the state-of-the-art.

**Compliance With Llm Reviewing Policy:**

Affirmed.

**Final Justification:**

The rebuttal addressed some of my concerns and clarified several aspects, but my overall assessment about the paper remains unchanged at this time.

**Key Questions For Authors:**

1. What's the gap between simulated numbers and profiled numbers?
2. What's the intellectual gap between the many prior works you've mentioned in related work vs this work, except for putting multiple works together under a bigger umbrella?

**Limitations:**

No. Please consider using profiling instead of simulation, which will force you to focus on reducing the profiling cost problem. Relying on simulation allows you to search through a large space with immunity.

**Strengths And Weaknesses:**

The paper addresses an important albeit well-explored one. The key innovation can be described as "prior work have done P-D disaggregation, A-F disaggregation, and considered heterogeneous resources, but we do all three simulataneously." It's quite incremental in that sense.

The key weakness of the paper is relying on simulation for the difficult part. Realistic numbers can only be obtained by profiling, and profiling is expensive. So reducing profiling cost is the crux of the problem, which the paper sidesteps.

Having said these, the paper does what it did pretty well.

Presentation and writing can be further improved. It's currently quite dense even though the ideas are simple enough.

---

> ### Author Rebuttal · Authors · 2026-03-31
>
> We thank the reviewer for these insightful questions and suggestions.
> ___
>
> > W1 & Q2. Incremental combination of prior work.
>
> We respectfully note that our contribution is not merely incremental. Our work addresses a fundamental bottleneck that emerges only when these techniques interact: traditional autoscaling leads to severe resource stranding. As shown in Figure 1, workload shifts impact operations drastically differently—incorporating 30% long requests causes attention load to surge 12× while FFN rises only 1.5×. Autoscaling a non-disaggregated architecture forces all components to scale together, massively over-provisioning some operations just to satisfy another's bottleneck. We observe that full disaggregation is fundamentally required to break this monolithic dependency, enabling independent, fine-grained autoscaling per inference phase to eliminate resource stranding. This is not a simple union of existing techniques—it is a synergy where AFD inherently requires autoscaling and autoscaling inherently benefits from AFD, achieving 1+1>2.
>
> This unification also introduces novel challenges absent in any prior isolated approach:
>
> (i) Architecturally, we demonstrate that fine-grained disaggregation with independent autoscaling achieves the best performance—neither monolithic scaling nor non-autoscaling baselines can address divergent Attention/FFN loads without waste. Compared with these baselines, our AFD-enabled autoscaling achieves up to 78.3% higher cost-efficiency (Figure 6), confirming that full disaggregation combined with autoscaling is essential for cost-efficient serving.
>
> (ii) Algorithmically, the 3-stage disaggregated pipeline on heterogeneous GPUs creates a highly complex NP-hard scheduling problem not present in prior 2-stage or homogeneous setups; our hierarchical framework with guided generation policy uniquely addresses this while reducing scheduling time by up to 78.2% versus exhaustive search with comparable throughput (Figure 10).
>
> (iii) Operationally, traditional global metrics (e.g., end-to-end latency) fail to isolate bottlenecks in a 3-stage pipeline; our phase-specific throughput pressure (E_t) enables precise, lag-free, independent scaling for each operation—validated by our metric characterization study (Figure 3).
>
> We will clarify these intellectual gaps more explicitly in the revision.
>
> > Q1. Gap between simulated and profiled numbers.
>
> We validated the simulator against real throughput for Qwen3-30B-A3B across five parallelism configurations on H100 GPUs. Absolute errors are 23–27% (e.g., dp=4/tp=1/ep=1: real 35.61 vs simulated 26.00, 27.0% error; dp=2/tp=2/ep=2: real 23.07 vs simulated 17.64 req/s, 23.5% error), primarily from accumulated decode-time estimation errors across autoregressive steps. Crucially, the simulator achieves perfect rank preservation across all five configurations, with normalized errors ≤5.5% (e.g., dp=4/tp=1/ep=1: real 1.000 vs simulated 1.000, 0.0% error; dp=2/tp=2/ep=2: real 0.648 vs. simulated 0.678, 4.6% error). Since our scheduling relies on relative throughput comparisons between candidates (§4.2.2) rather than exact absolute predictions, this relative accuracy suffices for correct scheduling decisions. We will include full validation results in the revision.
>
> > L1. Use profiling instead of simulation.
>
> We provide three pieces of evidence that simulation is both sufficient and preferable in our setting.
>
> First, empirical validation: we completely replaced simulation with physical hardware profiling for each candidate plan on OpenR1-Math and OpenThoughts workloads, keeping all settings identical to Figure 5. The deployment plans remained largely unchanged (only minor tp/dp swaps), and end-to-end performance differs by at most 4.3% (OpenR1-Math: 717 vs. 729 per-cost throughput, 1.7% gap; OpenThoughts: 398 vs. 415 per-cost throughput, 4.3% gap). This confirms our simulator produces nearly identical scheduling decisions to real profiling.
>
> Second, established methodology: simulation-based scheduling is widely adopted in state-of-the-art LLM serving—DistServe [OSDI'24], AlpaServe [OSDI'23], and MoE-Lightning [ASPLOS'25] all rely on similar approaches, validating that simulation provides sufficient fidelity for scheduling decisions. Profiling every candidate on real hardware would require orders of magnitude more GPU-hours, making scheduling itself a significant cost bottleneck.
>
> Third, and most critically for our system: during autoscaling, unseen workload patterns require re-evaluating deployment plans online. A profiling-based approach would execute each candidate on real hardware before selecting a new plan, introducing significant serving downtime—directly undermining the responsiveness autoscaling is designed to provide. Simulation avoids this bottleneck while maintaining fidelity, as our empirical comparison confirms.
>
> ___
>
> We will incorporate the additional experiments and analysis from our rebuttal into the revised paper.

---

> > ### Author Rebuttal · Reviewer_wvZ4 · 2026-04-01
> >
> > Thanks for the detailed response. There are already production systems like vLLM-Omni that can disaggregate omni models with many more than three stages. So the claims of difficulty and novelty of the three-stage problem is still somewhat incremental from the two-stage disaggregation people were doing before. I would've been completely convinced if your system could support arbitrary number of stages like vLLM-Omni.
> >
> > I also can't reconcile between statement in the effect of "profiling is infeasible" and then you profiling and running experiments for two whole datasets in less than a week.
> >
> > Overall, I'm slightly more positive but not enough to move a whole point.

---

> > > ### Author Response · Authors · 2026-04-01
> > >
> > > > Compare with vLLM-Omni.
> > >
> > > We thank the reviewer for the active engagement and thoughtful follow-up. We want to gently point out that this comparison conflates two distinct paradigms of disaggregation at essentially different granularities.
> > >
> > >
> > > * vLLM-Omni disaggregates the end-to-end multimodal service **across different model components** (e.g., Thinker LLM → Talker LLM → DiT Vocoder), orchestrating data flow between heterogeneous models in a multi-stage pipeline. Each component is a different model with different architectures. It's essentially a **multi-model serving orchestrator**.
> > > * HexGen-3 disaggregates **within a single model's inference**, separating prefill, decode-attention, and decode-FFN operations onto independent GPU pools. Each pool runs the same model weights but handles a different operation. It's a **single-model resource optimization** problem.
> > >
> > > Note that our work is *orthogonal* and *complementary* to vLLM-Omni.
> > > - For one thing, our core contribution is hierarchical scheduling that jointly optimizes resource allocation and parallel inference strategy layout per operation type across heterogeneous GPUs, plus fine-grained per-operation autoscaling. Our approach requires accommodating different communication paradigm within each operation (e.g., DP, TP, EP), and between different operations (e.g., transferring KV cache from prefill operation to decode-attention operation). These are not handled in vLLM-Omni's model-level disaggregation.
> > > - For another, our work can be applied to the auto-regressive LM component in vLLM-Omni's pipeline. Thus, the two systems are complementary rather than substitutive.
> > >
> > >
> > > > Profiling clarification.
> > >
> > > We appreciate the follow-up and would like to clarify the distinction.
> > >
> > > The profiling we conducted for rebuttal validation involved evaluating candidate plans on **two fixed, known workload traces offline** — this is feasible precisely because the workload is predetermined and we can afford hours of GPU time outside of any serving loop.
> > >
> > > The scenario where profiling becomes problematic is **online autoscaling**. When a previously unseen workload pattern triggers a scaling event during live serving, the scheduler must re-evaluate candidate deployment plans **in real time**. Profiling each candidate on real hardware at that point would impose minutes-to-hours of latency *within the serving path*, during which requests are either dropped or served suboptimally. Simulation bridges this gap by providing sufficient fidelity with orders-of-magnitude lower latency, enabling responsive online rescheduling — which is precisely the core use case of our system.
> > >
> > > ___
> > >
> > > We will integrate these clarifications and discussions into the revised manuscript to better clarify these distinctions. Thank you again for the valuable feedback.

---

### Official Review · Reviewer_ZmBr · 2026-03-08

**Soundness:** 3
**Presentation:** 3
**Significance:** 2
**Originality:** 2
**Overall Recommendation:** 3
**Confidence:** 5

**Summary:**

The paper introduces HEXGEN-3, an LLM serving framework designed to reduce operational costs by addressing the extreme workload heterogeneity inherent in production traffic. It achieves this by combining a fully disaggregated architecture—which separates prefill, attention, and Feed-Forward Network (FFN) operations into dedicated worker pools—with fine-grained heterogeneous resource autoscaling. The system operates on two synergistic pillars: a hierarchical scheduling framework that optimizes resource allocation and parallelism strategies for static setups , and an autoscaling framework that monitors real-time throughput metrics to dynamically adjust resources and reschedule deployments during workload fluctuations. Extensive evaluations show that HEXGEN-3 significantly improves per-cost throughput by up to 60% compared to state-of-the-art baselines under static provisioning, and up to 78.3% with autoscaling enabled under dynamic workloads.

**Compliance With Llm Reviewing Policy:**

Affirmed.

**Key Questions For Authors:**

1. In the homogeneous setup, SGLang exhibits significantly lower Time-Between-Tokens (TBT) latency than HEXGEN-3, yet it suffers from drastically worse Time-To-First-Token (TTFT) latency, which ultimately degrades its overall performance. It is unclear if the parallelism configuration for SGLang was rigorously tuned to find an optimal TTFT/TBT tradeoff, or if a default configuration was used that artificially handicapped the baseline.
2. Scheduling Scalability: Figure 10 indicates that the scheduling and rescheduling times run into the tens of seconds (and much higher without optimizations) for a relatively small 30B model on a small GPU cluster. This raises concerns about the framework's scalability when calculating optimization spaces for 70B+ models across hundreds of GPUs in real-time.
3. Evaluation Scale and Network Hardware: The evaluation utilizes Qwen3-30B-A3B on a cluster of 16-24 GPUs. Could you provide the specific intra-node and inter-node network bandwidths of your test environment? Furthermore, do you have experimental or simulated data showing how the AFD communication overhead scales with larger models (e.g., 70B+) or larger clusters where cross-node interconnects become the primary bottleneck?
4. Could you provide more details on the dynamic workload traces? How does the arrival rate change?
5. Simulator Accuracy: The local scheduler relies heavily on the AFD simulator, which assumes deterministic hardware performance and models network contention strictly via bandwidth-based transfer delays. What is the empirical error margin or accuracy of this simulator when compared against actual profiling metrics from real-world, high-contention deployments?
6. Coud you elaborate more on the system implementation for the AFD structure, like the request scheduler and batching, cudagraph.

**Limitations:**

yes.

**Strengths And Weaknesses:**

Strength

1. Addresses an Important Problem: The work tackles the prohibitively high operational costs of serving large language models (LLMs), which is driven by extreme workload heterogeneity in real-world production traffic.
2. Technically Sound Methodology: The paper proposes a rigorous hierarchical scheduling framework. This includes a global scheduler modeled as a Markov Decision Process solved via Simulated Annealing to handle resource allocation , and a local scheduler that optimizes parallelism configurations through constrained enumeration. Additionally, it uses a well-justified autoscaling framework driven by real-time throughput metrics.
3. Well-Structured and Clearly Written: The narrative is logical and easy to follow, clearly stating its three main contributions: the scheduling framework, the autoscaling framework, and the system implementation/evaluation. It flows cleanly from the system overview down into specific component details.

Weakness

1. Limited Evaluation Scale: The empirical evaluation is somewhat constrained in scope. The authors use a relatively small model (Qwen3-30B-A3B) for their benchmarks. Furthermore, the experimental hardware scale is limited to 16 to 24 GPUs (e.g., 16 H100s for homogeneous, and 8 H100s + 16 H20s for heterogeneous setups), which may not fully expose the bottlenecks of a fully disaggregated architecture in massive, multi-node data centers.
2. Scalability Concerns: The scheduling framework introduces significant overhead as the cluster size grows. Figure 10 illustrates that scheduling and rescheduling times run into the tens of seconds—and up to hundreds of seconds without optimizations—even for small clusters of up to 64 GPUs. This raises concerns about the system's ability to swiftly autoscale real-time deployments for hundreds of billions parameter models across hundreds to throusands of heterogeneous GPUs.
3. Lack of Experimental Details: Critical information needed to fully interpret the results is missing. The hardware setup completely omits intra-node and inter-node network bandwidth specifications , which are essential parameters for Attention-FFN Disaggregation (AFD) serving where communication overhead is a primary factor. Additionally, the paper lacks transparency regarding how the SGLang baseline was tuned for its parallelism strategy , and provides very little detail on how the dynamic workload traces were specifically generated beyond a high-level table.

---

> ### Author Rebuttal · Authors · 2026-03-31
>
> We thank the reviewer for these insightful questions and suggestions.
>
> > W1 & Q3. Limited evaluation scale.
>
> We add experiments on Mixtral 8x22B with 48 H100 GPUs. Results show HexGen-3 maintains cost-efficiency advantages at larger scale: 23.9% on NuminaMath CoT (7924.38 → 9821.4 token/s), 15.7% on OpenR1-Math (7765.98 → 8983.5 token/s), 14.2% on OpenThoughts (6676.77 → 7622.4 token/s) improvement in throughput over SGLang with 48 GPUs. These confirm our scheduling and disaggregation benefits generalize beyond small models.
>
> > W2 & Q2. Scheduling scalability concerns.
>
> First, model size does not increase scheduling complexity—it is an input parameter to the simulator, not a factor expanding the search space. The search space is determined by GPU count, GPU types, parallelism configurations, and worker types.
>
> Second, for large-scale clusters, we employ two complementary strategies:
>
> (1) Subset replication: identify common GPU subsets across heterogeneous resources and replicate optimized deployment plans across identical subsets, avoiding redundant per-subset search.
>
> (2) Parallel local search: since the local scheduler (§4.2.2) evaluates each candidate independently, we can parallelize these evaluations across CPU threads. We validate on a 128-GPU heterogeneous cluster: with 16 CPU threads, scheduling completes in 22s (10.1× speedup over single-threaded at 221s). This demonstrates HexGen-3 can swiftly autoscale hundred-billion-parameter models across large heterogeneous clusters without significant service disruption.
>
> > W3 & Q4. Missing experimental details.
>
> Network bandwidth: our setup uses 400 GBps NVLink (intra-node) and 50 GBps InfiniBand (inter-node). As shown in Figure 9, AFD communication latency accounts for only 13.6% of per-token generation latency on average, validating that communication does not bottleneck our architecture.
>
> SGLang tuning: we performed grid search over tensor parallelism degrees ∈{2,4,8}, data parallelism replicas, and prefill-to-decoding worker ratios, selecting the configuration maximizing throughput under the same resource budget. The optimal configuration: prefill workers with dp=4, tp=1; decoding workers with dp=6, tp=2.
>
> Trace generation: for each hour in Table 1, we subsampled from the corresponding dataset (WildGPT, OpenThoughts, OpenR1-Math, or NuminaMath), inheriting native input/output length distributions. Arrivals follow a Poisson process with rate λ=request_load/3600. The trace covers ~20× load variation (6.6K–129K req/hour) with alternating workload types to stress-test autoscaling under both load and distribution shifts.
>
> > Q1. SGLang TTFT/TBT tradeoff and baseline tuning.
>
> SGLang was rigorously tuned via grid search as described above (W3&Q4). The higher TBT in HexGen-3 is an inherent characteristic of AFD, not unfair tuning. AFD enables larger decoding batch sizes (2.2× on average vs. SGLang, Figure 12), and the alternating attention-FFN micro-batch computation pattern introduces additional per-token overhead—both contributing to increased TBT. However, this tradeoff significantly improves overall decoding throughput. This is a well-documented characteristic of AFD systems—MegaScale-Infer (Zhu et al., 2025, Figures 8–9) and Model-Attention Disaggregation (Chen et al., 2025, Figure 10) report the same pattern.
>
> Conversely, HexGen-3's TTFT is substantially lower because AFD's higher throughput reduces request queuing time. Despite the increased TBT, HexGen-3 achieves 16.3% average reduction in end-to-end latency vs. SGLang (Figure 12), confirming that throughput gains from AFD outweigh the TBT increase.
>
> > Q5. Simulator accuracy.
>
> We validated the simulator against real throughput for Qwen3-30B-A3B across five parallelism configurations on H100 GPUs. Absolute errors are 23–27% (e.g., dp=4/tp=1/ep=1: real 35.61 vs simulated 26.00, 27.0% error; dp=2/tp=2/ep=2: real 23.07 vs simulated 17.64 req/s, 23.5% error), primarily from accumulated decode-time estimation errors across autoregressive steps. Crucially, the simulator achieves perfect rank preservation across all five configurations, with normalized errors ≤5.5% (e.g., dp=4/tp=1/ep=1: real 1.000 vs simulated 1.000, 0.0% error; dp=2/tp=2/ep=2: real 0.648 vs. simulated 0.678, 4.6% error). Since our scheduling relies on relative throughput comparisons between candidates (§4.2.2) rather than exact absolute predictions, this relative accuracy suffices for correct scheduling decisions. We will include full validation results in the revision.
>
> > Q6. AFD system implementation details.
>
> The load balancer uses random bootstrap_room assignment, routing requests to prefill workers via bootstrap_room % dp_size. Decoding employs continuous batching with waiting/running queues and FCFS scheduling for new requests. With AFD enabled, the attention side retains continuous batching and FCFS, while FFN workers follow the attention workers' computation schedule—processing micro-batches as they arrive from attention workers.

---

> > ### Author Rebuttal · Reviewer_ZmBr · 2026-04-03
> >
> > LLM serving systems are commonly evaluated under a time-per-output-token (TPOT) SLO constraint. It would strengthen the evaluation to include a direct comparison between HexGen and SGLang under an common SLO setting (say 50ms), as this would provide a more controlled basis for assessing relative performance.

---

> > > ### Author Response · Authors · 2026-04-06
> > >
> > > > LLM serving systems are commonly evaluated under a time-per-output-token (TPOT) SLO constraint. It would strengthen the evaluation to include a direct comparison between HexGen and SGLang under a common SLO setting (say 50ms), as this would provide a more controlled basis for assessing relative performance.
> > >
> > > Thank you for this valuable suggestion. We agree that evaluating under a common TPOT SLO constraint could provide a more controlled and practically meaningful basis for comparison. We will integrate this as an essential case study in the revised paper.
> > >
> > > Following this suggestion, we conduct further experiments on an 8×H100 GPU cluster, comparing SGLang and HexGen-3 under a TPOT SLO constraint of 50ms. With this TPOT SLO constraint, the scheduling algorithm is directed toward a different optimization objective: it prunes all candidate solutions whose TPOT exceeds 50ms and searches for the configuration that maximizes throughput among the remaining feasible candidates.
> > >
> > > The results are shown in the following table. Under the 50ms TPOT SLO, HexGen-3 achieves 10–20% higher throughput than SGLang across all four workloads, while all configurations satisfy the SLO constraint with P95 ITL below 50ms.
> > >
> > > **Table 1: SLO-constrained throughput comparison**
> > >
> > > | System | Workload | Throughput (t/s) | Mean ITL (ms) | P50 ITL (ms) | P90 ITL (ms) | P95 ITL (ms) | Improvement |
> > > |--------|----------|------------|----------|---------|---------|---------|-------------|
> > > | SGLang | NuminaMath | 6,760 | 32.88 | 32.47 | 32.99 | 33.33 | — |
> > > | HexGen-3 | NuminaMath | 7,731 | 47.92 | 47.01 | 48.28 | 48.73 | +14.4% |
> > > | SGLang | OpenR1-Math | 6,554 | 33.26 | 33.20 | 34.05 | 34.43 | — |
> > > | HexGen-3 | OpenR1-Math | 7,632 | 46.63 | 46.44 | 47.21 | 47.87 | +16.5% |
> > > | SGLang | OpenThoughts | 4,995 | 35.56 | 34.69 | 39.95 | 42.12 | — |
> > > | HexGen-3 | OpenThoughts | 5,992 | 45.17 | 44.89 | 45.93 | 46.34 | +20.0% |
> > > | SGLang | WildGPT | 15,154 | 34.22 | 34.05 | 35.69 | 36.52 | — |
> > > | HexGen-3 | WildGPT | 16,687 | 48.34 | 48.05 | 49.32 | 49.41 | +10.1% |
> > >
> > > We report the optimal parallelism strategies selected by each system as follows:
> > > - **SGLang**: Prefill TP=2, Decoding DP=6
> > > - **HexGen-3**: Prefill TP=2, Attn DP=2, FFN DP=4
> > >
> > > Two key insights emerge from the strategy selection:
> > >
> > > - First, given the 50ms TPOT constraint, both SGLang and HexGen-3 prefer DP over TP for the decoding phase, as TP's inter-GPU synchronization overhead directly inflates per-token latency and would push TPOT beyond the SLO threshold.
> > > - Second, in HexGen-3, more resources are allocated to FFN workers (4 GPUs) than attention workers (2 GPUs) to keep the TPOT within the SLO, since FFN computation is the dominant contributor to per-token latency in the disaggregated decoding pipeline.
> > >
> > > We further validate the first insight by comparing two SGLang parallelism strategies on the same 8×H100 cluster, demonstrating that DP is more beneficial than TP under this SLO constraint:
> > >
> > > **Table 2: SGLang parallelism strategy comparison**
> > >
> > > | Strategy | Workload | Throughput (t/s) | Mean ITL (ms) | P50 ITL (ms) | P90 ITL (ms) | P95 ITL (ms) | SLO Met? |
> > > |----------|----------|------------|----------|---------|---------|---------|----------|
> > > | Prefill TP=2, Decoding DP=6 | NuminaMath | 6,760 | 32.88 | 32.47 | 32.99 | 33.33 | Yes |
> > > | Prefill TP=2, Decoding DP3×TP2 | NuminaMath | 4,811 | 49.37 | 49.25 | 53.84 | 55.52 | No |
> > > | Prefill TP=2, Decoding DP=6 | OpenR1-Math | 6,554 | 33.26 | 33.20 | 34.05 | 34.43 | Yes |
> > > | Prefill TP=2, Decoding DP3×TP2 | OpenR1-Math | 4,847 | 48.95 | 48.88 | 53.16 | 55.04 | No |
> > > | Prefill TP=2, Decoding DP=6 | OpenThoughts | 4,995 | 35.56 | 34.69 | 39.95 | 42.12 | Yes |
> > > | Prefill TP=2, Decoding DP3×TP2 | OpenThoughts | 3,624 | 49.59 | 49.48 | 53.73 | 55.36 | No |
> > > | Prefill TP=2, Decoding DP=6 | WildGPT | 15,154 | 34.22 | 34.05 | 35.69 | 36.52 | Yes |
> > > | Prefill TP=2, Decoding DP3×TP2 | WildGPT | 7,032 | 49.79 | 49.42 | 54.94 | 56.65 | No |
> > >
> > > The Decoding DP3×TP2 strategy violates the 50ms SLO across all workloads (P95 ITL: 55–57ms) while also achieving 25–55% lower throughput than Decoding DP=6. This confirms that under tight TPOT constraints, DP is the superior parallelism choice, as it avoids the fixed per-token communication overhead introduced by TP.
> > >
> > > We will include this analysis as a dedicated case study in the revised manuscript.

---

### Official Review · Reviewer_r7e5 · 2026-03-10

**Soundness:** 3
**Presentation:** 3
**Significance:** 2
**Originality:** 2
**Overall Recommendation:** 4
**Confidence:** 4

**Summary:**

This paper introduces HexGen-3, a disaggregated inference framework that separates workloads into three types: prefill, attention (decode), and FFN (decode). The system also targets computing platforms with heterogeneous hardware, which adds another layer of complexity to the scheduling problem. The system features two main parts: a scheduler and an autoscaler. The scheduler follows a two-step hierarchical design in which a global scheduler determines the best allocation policy based on the parallelism configuration policies produced by the local scheduler. For the autoscaler design, the authors claim that throughput metrics best align with workload fluctuations. Based on this, the system makes autoscaling decisions. The autoscaler is accelerated by a warm-start optimization. The system is implemented and evaluated on several workloads, demonstrating its superior performance in throughput per unit cost.

**Compliance With Llm Reviewing Policy:**

Affirmed.

**Final Justification:**

I would keep the weak accept assessment.

**Key Questions For Authors:**

- Your system relies heavily on the workload profile. In the paper, it is only briefly mentioned in a footnote what it comprises and how to obtain it (derived from a sliding window of recent traffic). More details are needed on how to reliably estimate the real-time workload. How do you handle workload spikes on short time scales?
- The simulator is an important part of the parallelism configuration algorithm. The paper does not seem to provide evaluation results on simulator accuracy. How well can it predict the performance of different components?
- The resource allocation is done at the granularity of worker type and GPU type. I was wondering whether this is necessary after all. If one GPU type definitely outperforms another for a particular worker type, why does it still make sense to use a mix of GPU types for that worker type?

**Limitations:**

Yes

**Strengths And Weaknesses:**

### Strengths

- The paper is well written and easy to follow, except that some of the figures have too small fonts that are hard to read.
- The paper captures multiple design points in a single system: prefill-decode disaggregation, attention-FFN disaggregation, heterogeneous hardware, resource allocation, parallelism configuration, and autoscaling. For each of these points, the paper proposes a reasonably good solution to it.

### Weaknesses

- The idea to address multiple design points in a single system is good and challenging. However, the proposed solution for each of these points does not seem innovative, nor does it provide any new insights. Resource allocation and autoscaling are both well-studied topics, and the disaggregation angle does not change the fundamental problem setup much.
- Figure 3 provides an essential observation for the design, yet it is based on a single production workload trace. It would be nice if the same observation were verified on other production traces.
- Throughout the paper, performance metrics like per-cost throughput and P99 latency cost are considered. In real-world deployments, inference workloads typically have a latency SLO, which is not mentioned in the paper. It is unclear how the proposed system can be adapted to SLO-oriented performance metrics.
- The profiling overhead of the system seems quite high. However, this aspect is not sufficiently covered in the paper.
- The scheduling time overhead is also quite high for large-scale systems. The paper mentions that the proposed optimizations can reduce this overhead, but it does not comment on how well the proposed system can keep up with real-world workload fluctuations.

---

> ### Author Rebuttal · Authors · 2026-03-31
>
> Thank you for your insightful questions and suggestions.
>
> > W1. System novelty.
>
> AFD fundamentally changes the problem setup. Attention and FFN exhibit very different load reactions to workload changes—e.g., incorporating 30% long requests causes attention load to surge 12× while FFN rises only 1.5× (Figure 1). Traditional autoscaling forces the entire decoding unit to scale together, causing severe resource stranding—over-provisioning FFN compute just to satisfy an attention memory bottleneck. Our core insight: AFD is not merely combining existing techniques—AFD inherently requires autoscaling to maintain efficiency under such divergent scaling behaviors, and autoscaling naturally enhances AFD's ability to handle fluctuating workloads. This synergy enables independent, fine-grained scaling based on real-time bottlenecks, achieving 1+1>2.
>
> > W2. Figure 3 based on single trace.
>
> We conducted additional analysis using the Azure production trace (Patel et al., 2024), measuring the Pearson correlation between throughput metrics and actual service load. The Azure trace yields 0.9135, while the Figure 3 trace yields 0.9227—both indicating strong linear correlation. These consistently high correlations across distinct production environments confirm throughput metrics reliably track workload changes, supporting their use as our primary autoscaling signal.
>
> > W3. Lack of SLO evaluation.
>
> We evaluated SLO attainment following AlpaServe [OSDI'23]'s method. Results show our system minimizes the latency to achieve 95% and 99% SLO attainment: 16.6%/30.0% reduction on NuminaMath CoT (Latency Cost 1.08/1.16 → 0.87/0.88), 19.9%/24.0% on WildGPT (Latency Cost 2.78/2.75 → 1.90/1.93).
>
> > W4. Profiling overhead.
>
> Our system uses a roofline-style simulator (Appendix A) instead of hardware profiling, analogous to MoE-Lightning [ASPLOS'25]. Per-iteration simulation time is only 0.3–0.9s (total scheduling times in Figure 10), orders of magnitude smaller than real hardware profiling which requires executing inference workloads on GPUs.
>
> > W5. Scheduling overhead vs. real-world fluctuations.
>
> Our scheduling overhead is already small for typical workload fluctuations and can be further minimized for faster changes. We analyze two cases:
>
> (1) For minute/hourly-scale changes (the common case in production), our rescheduling optimizations such as warm-start and cost-aware search (§5.2.2) keep overhead well within the fluctuation timescale—e.g., rescheduling completes in under 25s for 64-GPU clusters (Figure 10), negligible relative to hourly workload shifts.
>
> (2) For faster changes on the order of seconds, the search is fully parallelizable: with 16 CPU threads, rescheduling time on a 64-GPU cluster drops from ~25s to <10s, enabling the system to keep pace with rapid fluctuations.
>
> > Q1. Workload profile and spike handling.
>
> The workload profile W=⟨λ,μ_in,μ_out⟩ is estimated from a 300s sliding window of recent traffic, continuously aggregating request metadata (arrival timestamps, input/output token lengths). This approach is analogous to established frameworks like DistServe [OSDI'24]. As shown in our autoscaling evaluation (Figure 6), this proves effective, enabling HexGen-3 to achieve up to 38% improvement over non-autoscaling baselines under dynamic workloads. For spike handling, rather than relying solely on estimated profiles which may lag: (1) throughput-pressure monitoring triggers scaling immediately when E_t exceeds hysteresis thresholds (§5.1); (2) proportional scaling aligns capacity in a single step rather than slow incremental ramping (§5.2.1); (3) soft scaling-in reinstates draining resources during spikes, avoiding cold-start latency (Appendix D).
>
> > Q2. Simulator accuracy.
>
> We validated the simulator against real throughput for Qwen3-30B-A3B across five parallelism configurations on H100 GPUs. Absolute errors are 23–27%, primarily from accumulated decode-time estimation errors. Crucially, the simulator achieves perfect rank preservation across all configurations, with normalized errors ≤5.5%. Since our scheduling relies on relative throughput comparisons between candidates (§4.2.2) rather than exact predictions, this relative accuracy suffices for correct scheduling decisions. Full validation results will be in the updated manuscript.
>
> > Q3. Why mix GPU types per worker type?
>
> No single GPU type always outperforms another—the optimal hardware depends on parallelism configuration and workload. For example, H20 (96GB) can preserve more activation memory for larger FFN batches, while H100 offers higher compute for small batches. Under different workloads, the same worker type may prefer different GPUs, making mixed allocation beneficial.
>
> Moreover, resource constraints make mixing necessary: in a cluster with 8×H100+16×H20, exclusively assigning H100s to one worker type starves others. Our global scheduler (§4.2.1) redistributes to resolve bottlenecks—mixed allocation emerges from optimization, not imposition.

---

> > ### Author Rebuttal · Reviewer_r7e5 · 2026-04-01
> >
> > Thank you for your response to my questions. Most technical questions have been resolved. However, I still have some reservations about the novelty. I would be more positive about the paper if it were the first to propose AFD. However, this is unfortunately not the case. AFD has been discussed in previous works (e.g., MegaScale-Infer). I would keep my original score.

---

> > > ### Author Response · Authors · 2026-04-01
> > >
> > > We thank the reviewer for acknowledging the resolution of technical questions.
> > >
> > > We respectfully clarify that **AFD itself is not our claimed contribution**. Rather, we *identify* and *solve* the critical resource management challenges that arise when deploying disaggregated systems under dynamic, heterogeneous workloads.
> > >
> > > As our experiments demonstrate, since attention and FFN loads shift disproportionately under workload heterogeneity (e.g., varying long/short request ratios, as shown in Figure 1), naively applying AFD to fluctuating workloads can actually *degrade* performance compared to non-AFD systems. MegaScale-Infer, presented as one of our baselines, further exemplifies this point: It enables AFD but obtains even worse performance than SGLang (baseline non-AFD system) under fluctuating workloads.
> > >
> > > Making disaggregated systems work well under dynamic workloads is, we believe, also an essential and non-trivial point in promoting and informing future system research.
> > >
> > > We will integrate this clarification into the revised manuscript. Thank you again for the valuable feedback.

---

### Decision · Program_Chairs · 2026-04-30

**Decision:**

Accept (regular)

**Comment:**

HexGen-3 addresses LLM serving cost under dynamic heterogeneous workloads by combining attention-FFN disaggregation with fine-grained heterogeneous autoscaling, reporting strong empirical gains (up to 78.3% per-cost throughput improvement). Reviewers broadly agree the system is technically solid and well-evaluated, but two raise legitimate concerns about incremental novelty — AFD itself is not new, and the scheduling/autoscaling components are individually well-studied. The rebuttal effectively clarified that the core contribution is the synergy between AFD and autoscaling (which prior systems like MegaScale-Infer do not capture), added cross-model validation on Mixtral 8x22B, SLO-constrained evaluation, and convincingly justified the simulation-over-profiling choice for online rescheduling. With scores of 4/3/3/5, I lean toward acceptance given the practical significance and the quality of the rebuttal responses.